# Biophysical neural adaptation mechanisms enable artificial neural networks to capture dynamic retinal computation

Saad Idrees ●[1,2] ✉, Michael B. Manookin ●[3], Fred Rieke[4], Greg D. Field ●[5] & Joel Zylberberg ●[1,2,6] ✉

Adaptation is a universal aspect of neural systems that changes circuit computations to match prevailing inputs. These changes facilitate efficient encoding of sensory inputs while avoiding saturation. Conventional artificial neural networks (ANNs) have limited adaptive capabilities, hindering their ability to reliably predict neural output under dynamic input conditions. Can embedding neural adaptive mechanisms in ANNs improve their performance? To answer this question, we develop a new deep learning model of the retina that incorporates the biophysics of photoreceptor adaptation at the front-end of conventional convolutional neural networks (CNNs). These conventional CNNs build on 'Deep Retina,' a previously developed model of retinal ganglion cell (RGC) activity. CNNs that include this new photoreceptor layer outperform conventional CNN models at predicting male and female primate and rat RGC responses to naturalistic stimuli that include dynamic local intensity changes and large changes in the ambient illumination. These improved predictions result directly from adaptation within the phototransduction cascade. This research underscores the potential of embedding models of neural adaptation in ANNs and using them to determine how neural circuits manage the complexities of encoding natural inputs that are dynamic and span a large range of light levels.

Artificial neural networks (ANNs) combined with deep learning algorithms are useful in modeling the function of the nervous system and are being used to model and investigate many brain areas[1]. Under relatively controlled conditions, ANNs perform well in computer vision tasks such as object recognition[2–4], and they can successfully predict the responses of neurons in visual cortex[5–8] and retina[9–15]. However, it is less clear how ANNs perform in naturalistic settings, where, for example, the statistics of sensory input can vary significantly from moment to moment. A specific concern is that the static nonlinear functions that ANNs typically employ will limit their ability to

dynamically adapt to changing input conditions. Accounting for such dynamics is important because adaptation is a nearly universal feature of individual neurons and neural circuits[16,17].

Sensory systems provide some of the clearest examples of the importance of adaptation. For example, adaptation causes the fading of perceived intensity of odors[18] and accommodation to sounds[19]. Within the visual system, adaptation constantly adjusts neural responses to match the prevailing input conditions. During natural vision, the amount of light falling on the retina can change locally and globally by several orders of magnitude on timescales ranging from

[1]Department of Physics and Astronomy, York University, Toronto, ON, Canada. [2]Centre for Vision Research, York University, Toronto, ON, Canada. [3]Department of Ophthalmology, University of Washington, Seattle, WA, USA. [4]Department of Physiology and Biophysics, University of Washington, Seattle, WA, USA. [5]Stein Eye Institute, Department of Ophthalmology, University of California, Los Angeles, CA, USA. [6]Learning in Machines and Brains Program, Canadian Institute for Advanced Research, Toronto, ON, Canada. ✉e-mail: saidrees@yorku.ca; joelzy@yorku.ca

fractions of a second (e.g., eye movements such as saccades), to minutes (e.g., movement between sunlight and shade), and to hours (e.g., the rising and setting sun). The limited dynamic range of individual neurons makes adaptation essential to match the range of neural responses to the current range of inputs. In vision, much of this adaptation occurs in the retina, and as early as the photoreceptors. Indeed, phototransduction can adapt rapidly and dynamically to control the sensitivity and kinetics with which light inputs are converted into electrical signals[16,20–22].

We examined whether incorporating photoreceptor adaptation into ANNs enhanced their accuracy at predicting neural responses under varied input conditions. We tested the ability of a convolutional neural network (CNN), similar to Deep Retina[9], to predict stimulus-evoked retinal ganglion cell (RGC) firing patterns under lighting conditions that differed from those under which they were trained. CNNs failed to generalized to new lighting conditions, and this failure motivated us to create a new type of CNN layer that incorporates a biophysical model of photoreceptor adaptation[20,23] that emulates the transformation of light into electrical signals. The photoreceptor layer can be used as an input to conventional CNNs, and can be trained end-to-end along with the other CNN layers. It thus equips deep-learning CNN models of the retina with biorealistic adaptation mechanisms. We found these biophysical photoreceptor–CNN hybrid models better generalized across lighting conditions that were not included in training. Furthermore, because the photoreceptor adaptation was local, the photoreceptor–CNN model outperforms conventional CNN

models in tasks involving rapid changes in local light intensity. These results suggest that chimeric models blending biophysical realism with trainable CNNs are better at modeling neural activity. Moreover, they provide a promising direction for investigating how adaptation mechanisms shape neural circuit function.

## Results

### Photoreceptor adaptation improves CNN performance at predicting RGC responses to natural stimuli

We hypothesized that incorporating photoreceptor adaptation could improve the performance of CNN models at predicting RGC responses to naturalistic stimuli that involve local luminance variations. To test this hypothesis, we recorded the spiking activity of primate RGCs using a high-density multielectrode array (Methods), and then attempted to predict these visually-evoked responses using CNN models either with or without photoreceptor adaptation mechanisms. CNN training used measured responses to checkerboard noise and naturalistic movies (Methods and Fig. 1a). The checkerboard noise provided responses to statistically stable stimuli, while the naturalistic movies presented large and rapid changes in light intensity characteristic of the retinal input during natural vision. Both stimuli were presented at a mean luminance of $50\ R^*receptor^{-1}s^{-1}$, where RGC responses primarily depend on rod photoreceptor responses, and the rods are at a light level where their gain is adapting strongly to the stimulus[24,25]. We selected 57 RGCs (27 ON and 30 OFF parasol cells) for modeling purposes

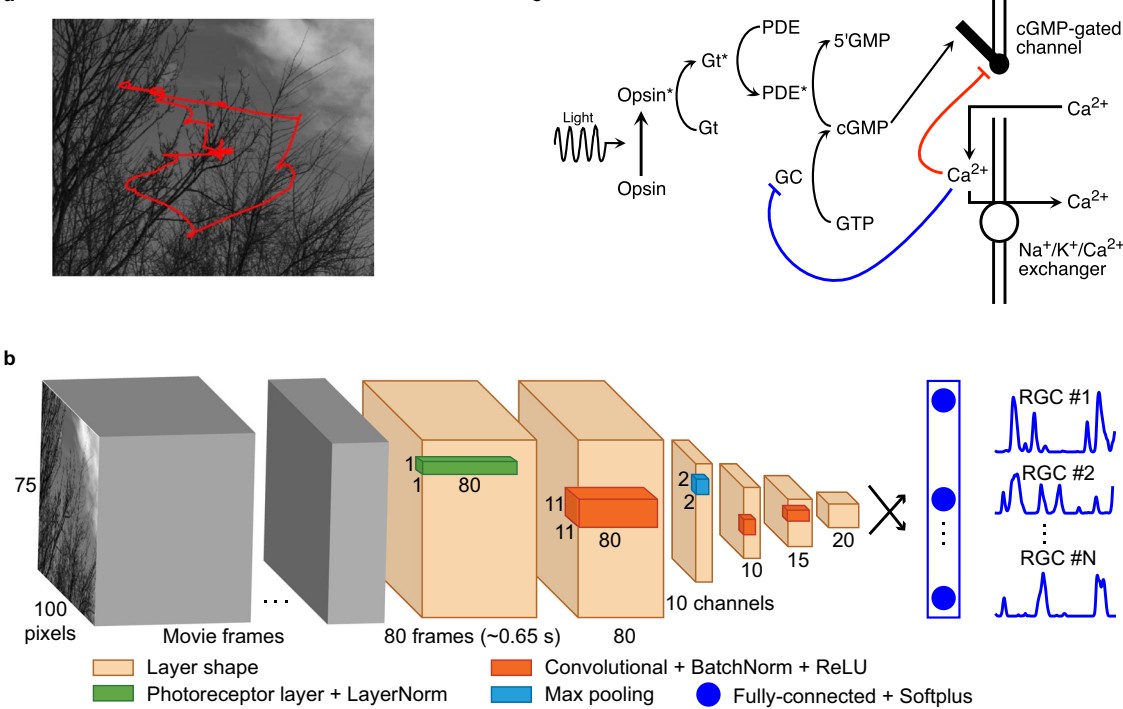

**Fig. 1 | Training and architecture of photoreceptor–CNN/conventional CNN model. a** Naturalistic movie generated by displacing natural scene images from Van Hateren dataset[54] across the retina, to mimic eye movement trajectories (red lines) derived from the DOVES dataset[55]. **b** Photoreceptor–CNN Model architecture incorporating a photoreceptor layer at the front-end (green) followed by Layer Normalization and 3 convolution layers (orange). The model output is a fully-connected layer that has N units based on the number of RGCs in the dataset followed by a softplus activation function that transforms the outputs into RGC spiking output (blue traces). By traversing through the input movie 80 frames at a time, an entire time series of RGC responses is obtained. When configured as a conventional CNN (without the photoreceptor layer), Layer Norm before the first convolution layer is the input layer. **c** Schematic showing the phototransduction

cascade and corresponding components of the biophysical model (adapted from ref. 20). Continuous synthesis of cGMP by guanylate cyclase (GC) opens cGMP-gated channels in the membrane. Activation of light-sensitive opsin (Opsin*) results in channel closure through the activation of G-protein transducin (Gt*), subsequently activating PDE* and decreasing cGMP concentration. Calcium ions ($Ca^{2+}$) enter the photoreceptor outer segment via cGMP-gated channels and are extruded through $Na^+/K^+/Ca^{2+}$ exchangers in the membrane. The biophysical model incorporates two distinct calcium-dependent feedback mechanisms influencing the rate of cGMP synthesis (blue line) and the activity of the cGMP-gated channels (red line). In the experiments presented here, the strength of Ca-dependent feedback to the cGMP-gated channels was set to zero, based on observations from rod photoreceptor cells[23]. See Supplementary Table 2 for a list of parameters and their values.

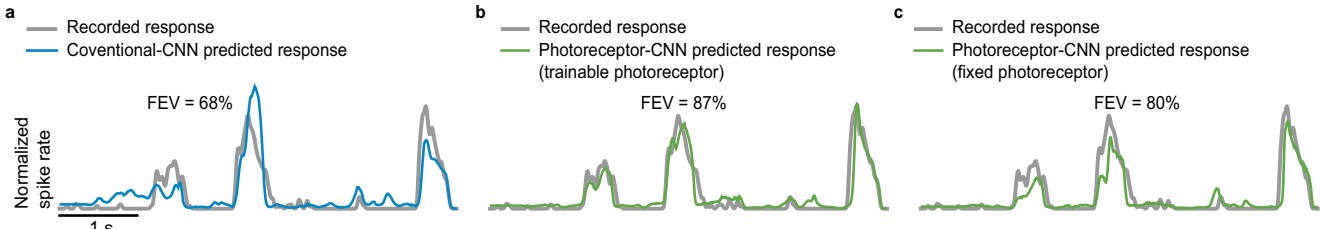

**Fig. 2 | Incorporating photoreceptor adaptation improves CNN performance in predicting an example RGC's response to naturalistic movies.** Recorded response of an example primate parasol RGC to a held-out naturalistic movie shown as normalized spike rate (gray), and predicted responses by (**a**) conventional CNN model (blue), (**b**) photoreceptor–CNN model with trainable photoreceptor layer (green), and (**c**) photoreceptor–CNN model with non-trainable photoreceptor layer parameters fixed to experimental fits to rods (Supplementary Table 2). Fraction of Explainable Variance Explained (FEV) values quantify the percentage of variance in the RGC's actual responses that could be explained by each model. Source data are provided as a Source Data file.

based on spike sorting quality and reliability across experimental conditions (Methods).

We constructed a conventional CNN model similar to the existing state-of-the-art Deep Retina architecture[9] (i.e., the architecture in Fig. 1b with the photoreceptor layer removed, Methods). The model cascades multiple convolutional layers and rectification to extract spatiotemporal information from the input. Each convolution is followed by Batch Normalization that z-scores the outputs (Methods) before rectification. A fully-connected layer at the end, followed by a softplus nonlinearity, maps the extracted features to the spiking activity of RGCs. We optimized model hyperparameters – including the number of layers and the number of channels in each layer–for our dataset (see Methods; Supplementary Table 1). The model takes as input a 640 ms movie segment and produces as output an instantaneous spike rate for each RGC at the end of that movie segment. In other words, the response at each time point was based on the previous 640 ms. This process was repeated every 8 ms to predict the entire time series of RGC responses evoked by the stimulus movie. A Layer Normalization (Layer Norm) layer at the input standardized the value of each pixel of the input movie segment at each time point relative to its mean value across the movie segment. This enabled the CNN to compensate for changes in input magnitude across light levels, ensured stability during training, and promoted faster convergence (Methods). Model performance was quantified using the fraction (expressed as a percentage) of explainable variance in the RGC responses that was captured by the model (FEV; see Methods). A perfect model, by definition, would yield an FEV value of 100%. Comparisons between models were facilitated by presenting median FEV values along with 95% confidence intervals.

Due to the limited duration of the naturalistic movies in our dataset, we could not fully train the models on the naturalistic movie data alone. Instead, we adopted a two-stage training process. First, we trained the model to predict RGC responses using the entire white noise movie dataset, totaling 36 min. Second, we fine-tuned the resulting model using RGC responses to eight of the nine naturalistic movies (8 min of the data) and evaluated the model on the held-out movie (6 s). For an example RGC, this conventional CNN model captured 68% FEV of the recorded (Fig. 2a) response with a median FEV of 68% ± 12% across the population ($N = 57$) of recorded cells (Fig. 3a).

To test the hypothesis that dynamic photoreceptor adaptation improves the CNN predictions, we developed a new type of CNN layer based on a biophysical model of the phototransduction cascade by ref. 20. We then built a new CNN model with this photoreceptor layer at its input and tested it with the same procedure used to test the conventional CNN (above). This photoreceptor model has previously been validated for its faithful representation of photoreceptor adaptation dynamics[20]. The model is based directly on the signaling cascade that constitutes the phototransduction process (Fig. 1c). Rapid adaptation in this model emerges primarily from changes in the rate of cGMP

turnover produced by light intensity-dependent changes in phosphodiesterase activity[26]. The biochemical reactions of the phototransduction cascade were represented by a set of six differential equations that also incorporate dynamic feedback mechanisms to the cGMP-gated channels as detailed in the biophysical model by ref. 20 (reproduced in Supplementary Note 2). Model parameters can be adjusted to match responses of either rod or cone photoreceptors (ref. 23; Supplementary Table 2). Here, we configured the photoreceptor model to represent primate rods as the underlying electrophysiology experiment was conducted at a mean luminance of 50 R*receptor$^{-1}$s$^{-1}$, where rod photoreceptors primarily drive RGC responses.

The fully trainable CNN layer encapsulating the biophysical photoreceptor model is characterized by twelve parameters (Methods) that could be trained together with the downstream network through backpropagation. This layer converts time-varying light intensity signals at each pixel in the input movie–measured in units of receptor activations per photoreceptor per second (R*receptor$^{-1}$s$^{-1}$)–into time-varying photocurrent values (measured in pA). In the hybrid biophysical photoreceptor–CNN model, the photoreceptor layer functions as the input stage of the CNN (Fig. 1b, photoreceptor layer). Following the photoreceptor layer, Layer Normalization is implemented to normalize the input distribution to the CNN, while preserving the temporal structure of the photocurrents. This design also ensures that the parameters of the biophysical model, having a different scale than the downstream CNN weights, can be trained together with the CNN through backpropagation. Henceforth, we refer to this hybrid model as the photoreceptor–CNN model. Unless explicitly stated, we allow some of the photoreceptor model parameters to be learned along with downstream CNN weights, and hence to deviate from the values to which they are initialized (in this case rod photoreceptor parameter values). A description of the trainable and non-trainable parameters is provided in the "Methods" section.

Similar to the conventional CNN, the photoreceptor–CNN model (with optimized hyperparameters; Supplementary Table 1) was trained to predict primate RGC responses first to a white noise movie and then model weights were tuned by training to naturalistic movies. When evaluated on the same held-out movie segment as the conventional CNN, the predicted responses of an example RGC generated by the photoreceptor–CNN model (Fig. 2b) were much more closely matched to the actual response than the predictions from the conventional CNN (Fig. 2b). Overall, the photoreceptor–CNN model performed with a median FEV of 81% ± 6% (Fig. 3a). This performance gain is substantial given that the photoreceptor layer enhanced the predictive capability of conventional CNNs by approximately 19% ($p = 1 \times 10^{-7}$, two-sided Wilcoxon signed-rank test, $N = 57$ RGCs). The enhanced capability of the photoreceptor–CNN model was robust when trained and evaluated over different combinations of naturalistic movies (Supplementary Fig. 1) and underscores the pivotal role played by the simulated photoreceptor layer.

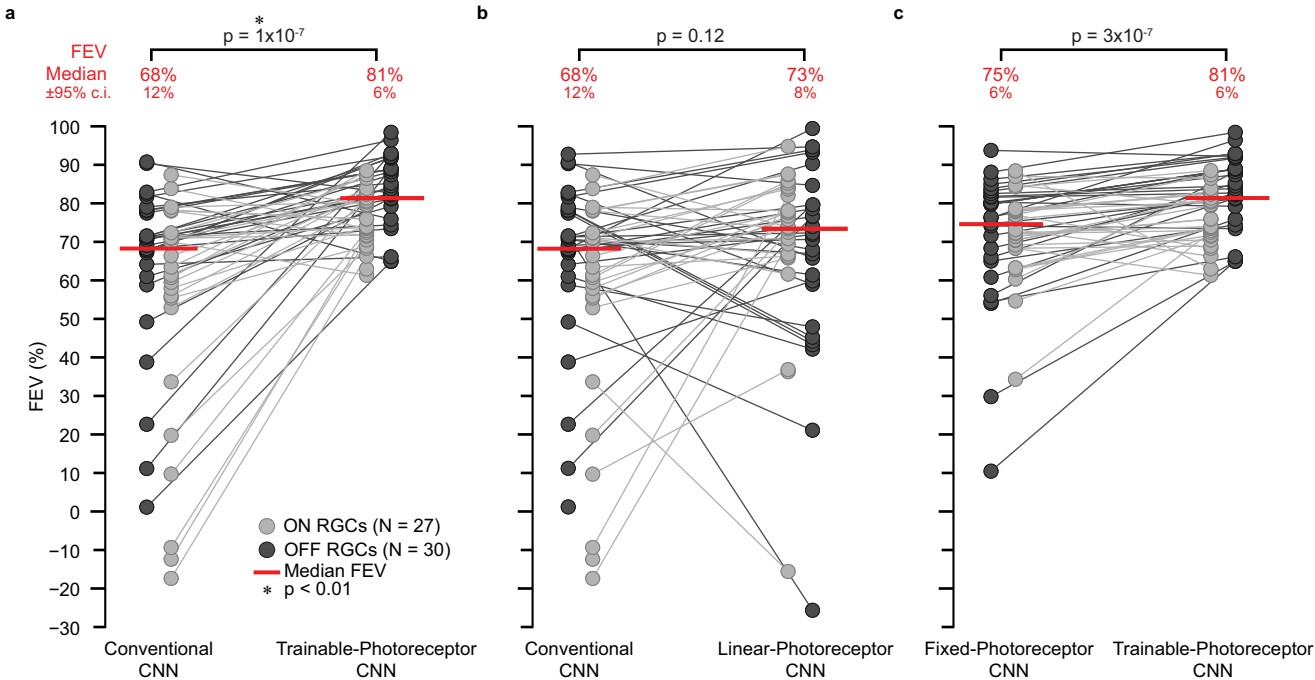

**Fig. 3 | Incorporating photoreceptor adaptation improves CNN performance in predicting RGC responses to naturalistic movies. a** Comparison between conventional CNN and photoreceptor–CNN model with the photoreceptor layer parameters trained with the downstream CNN. Y-axis shows the performance of conventional CNN (left) and photoreceptor–CNN model (right) as FEV values for each RGC (circles). Light gray circles denote ON type RGCs (*N* = 27), and dark gray circles denote OFF type RGCs (N = 30). Connecting lines link the FEV values for each RGC across models. Median FEV values across all RGCs (*N* = 57) are indicated by red lines, and stated as FEV ± 95%c.i. in red text at the top. P-values were calculated by performing paired two-sided Wilcoxon signed-rank test on the FEV distributions

from the CNN and photoreceptor–CNN model. An asterisk indicates statistically significant difference (*p* < 0.01) between performance of the two models. **b** Similar comparisons as in (**a**) but between conventional CNN (same as **a**, left) and photoreceptor–CNN, with the biophysical photoreceptor model being replaced by a linear empirical photoreceptor model. **c** Similar comparison as in **a** between photoreceptor–CNN models with the biophysical photoreceptor layer parameters fixed to experimental fits to primate rods (left; Supplementary Table 2) and where the photoreceptor layer parameters were learned along with the downstream CNN (right; same as **a**, right). Source data are provided as a Source Data file.

The superior performance of the photoreceptor–CNN model can be attributed to its ability to capture and leverage dynamic photoreceptor adaptation. At the ambient light level of this experiment (50 R*receptor$^{-1}$s$^{-1}$) rod photoreceptors are adapting strongly[24,25]. Conventional CNNs, despite incorporating Layer Norm at their input to accommodate steady-state sensitivity changes associated with the mean intensity of the stimuli, struggle to capture this adaptation. Photoreceptor–CNNs explicitly model this dynamic adaptation, making them more effective in predicting RGC responses in this setting.

**Adaptation in the photoreceptor layer drives performance gains**
What causes the photoreceptor–CNN to outperform the conventional CNN at predicting RGC responses? Notably the superior performance is not attributable to an increase in model capacity from the addition of 12 trainable parameters of the photoreceptor layer. In fact, the conventional CNN had to be much larger (873,642 parameters) without the photoreceptor layer (538,107 parameters) to achieve its ceiling performance. This difference in parameter count resulted from separately optimizing the size of CNN and photoreceptor–CNN models to best predict responses via grid searches over the model hyperparameters (Methods; Supplementary Note 1).

Given the inherent limitation of CNNs in dynamically adapting based on the prevailing inputs, we hypothesized that the observed performance gains in the photoreceptor–CNN model stem from the adaptation mechanisms embedded in the photoreceptor layer that dynamically adjust response sensitivity and kinetics based on recent stimulus history. To test this hypothesis, we substituted the nonlinear biophysical photoreceptor model with an empirical linear photoreceptor model[20] (Methods). This linear model consists of a linear filter governed by five trainable parameters. These parameters were

initialized based on experimentally measured single-photon responses[20]. Unlike the biophysical model, the linear photoreceptor model lacks the ability to dynamically adjust its sensitivity. In this model, a single parameter provides sensitivity adjustment, applied to the entire linear photoreceptor model output to account for adaptation.

Following the same procedure as for the conventional CNN and the photoreceptor–CNN, the hyperparameters of the linear photoreceptor–CNN model were optimized via a grid search (Methods; Supplementary Table 1). The resulting model was first trained (including the 5 photoreceptor parameters that were initialized to experimental values) to predict RGC responses to the white noise movie and then fine-tuned using RGC responses to the naturalistic movies. When evaluated on the held-out segment of naturalistic movie data, this model performed very similarly to the conventional CNN model, yielding FEV of 73% ± 8% (Fig. 3b; *p* = 0.12, two-sided Wilcoxon signed-rank test, *N* = 57 RGCs). This is expected since the initial linear filtering stages of the conventional CNN should be able to capture the filtering performed by the simplified linear photoreceptor model. Taken together, these experiments underscore the significance of adaptation as crucial components influencing model predictions.

We also tested photoreceptor–CNN models in which the biophysical model parameters were non-trainable, ensuring that they did not deviate from their experimental fits to rods (Supplementary Table 2) during training. As a result, the photoreceptor layer in the trained model represented true biological rods. The resultant photoreceptor–CNN model trained on the same task as above, exhibited lower performance than its fully-trainable counterpart (Fig. 2c; Fig. 3c, FEV 75% ± 6%, *p* = 3 × 10$^{-7}$ two-sided Wilcoxon signed-rank test, *N* = 57 RGCs) but still outperformed the conventional-CNN model when evaluated on the

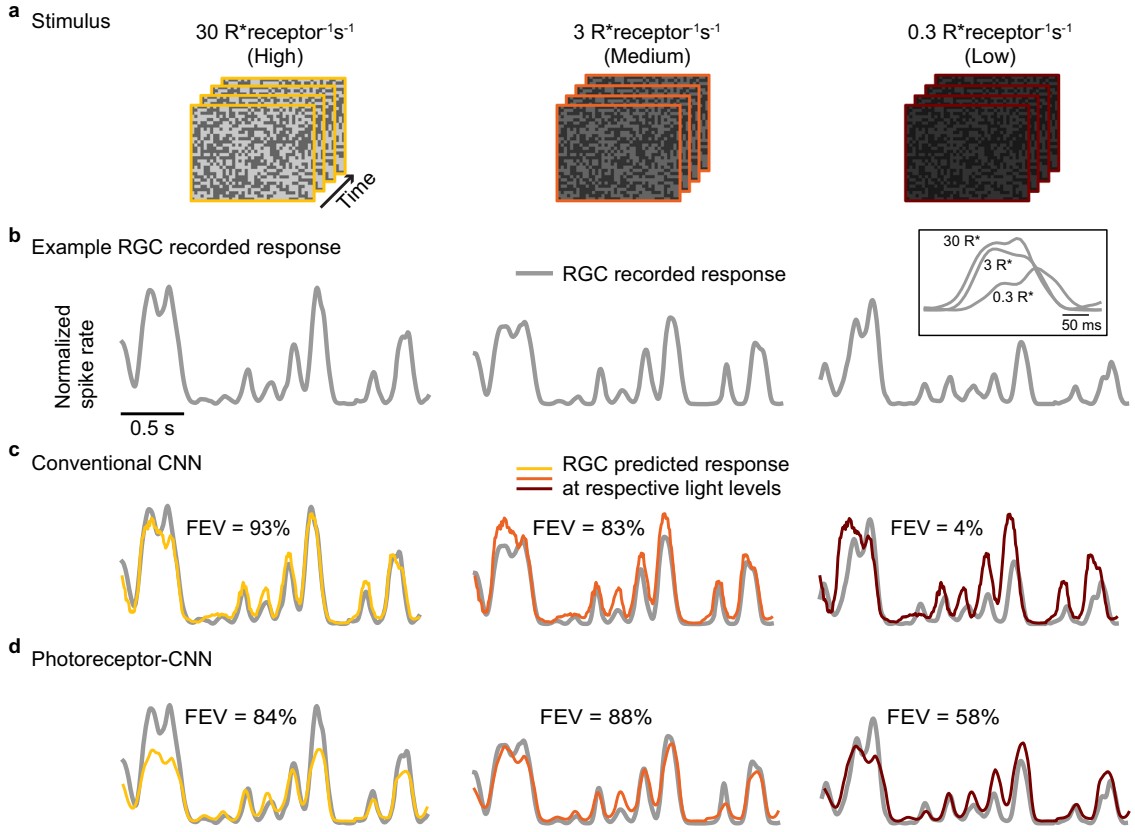

**Fig. 4 | Incorporating photoreceptor adaptation enables CNN to predict responses of an example RGC at a light level different from those at which it was trained. a** White noise movie at three different light levels: high (column 1; yellow), medium (column 2; orange) and low (column 3; red). **b** Recorded response (normalized spike rate) of an example RGC (gray lines) to white noise movie at the three different light levels in (**a**) (columns). Inset above the right column overlays a segment of the responses at the three light levels to directly compare response kinetics. **c** Responses predicted by a conventional CNN model (colored) at each light level in (**a**) (columns). FEV values above each trace quantify the performance of the model for this RGC at the corresponding light levels. **d** Same as in **c** but for the proposed photoreceptor–CNN model. Models were trained on data at high 30 R*receptor⁻¹s⁻¹ (column 1) and medium 3 R*receptor⁻¹s⁻¹ (column 2) and evaluated at low 0.3 R*receptor⁻¹s⁻¹ (column 3) light level. Source data are provided as a Source Data file.

naturalistic movie dataset. These findings further suggest that the observed performance gains can indeed be attributed to the nonlinear properties such as adaptation in the photoreceptors.

## Incorporating photoreceptor adaptation enables CNNs to generalize across light levels

The results so far indicate that the photoreceptor–CNN model shows superior performance in predicting responses to naturalistic movies that include dynamic local changes in intensity but lack changes in global light level. In the following sections, we shift our focus to asking how well the models generalize over global changes in mean light level, reminiscent of natural vision but occurring at slower time scales. This entails training the models at two distinct light levels and evaluating their performance on a third level not encountered during training.

To generate the experimental data for these computational studies, we recorded the spiking activity of primate RGCs to binary checkerboard noise movies (Methods) presented at three different light levels (30 R*receptor⁻¹s⁻¹ (high), 3 R*receptor⁻¹s⁻¹ (medium) and 0.3 R*receptor⁻¹s⁻¹ (low)). Rod photoreceptors dominate signaling at all three light levels. At 30 R*receptor⁻¹s⁻¹ the gain and kinetics of rod responses adapt strongly[24,25], while at 0.3 R*receptor⁻¹s⁻¹ rod responses adapt minimally because of the low rate of photon arrival at an individual rod. RGC responses to the checkerboard noise were recorded for 60 min at each light level. The analysis focused on 37 RGCs based on spike sorting quality and reliably tracking cells across the light levels (Methods).

The conventional CNN architecture (same as the one used above) was re-optimized for the numbers of convolutional layers, filters in each layer, and the filter sizes with a grid search on this new dataset (Supplementary Table 1). The model takes as input a 996 ms movie segment. We chose a longer segment length in these experiments to account for longer integration times at the lower light levels.

This CNN model was trained to predict RGC responses to a total of 40 min of a checkerboard noise presented at high and medium light levels. We evaluated model performance using a test data set that included 5–10 s of held-out segments of the checkerboard noise movie at all light levels (Fig. 4a), including the low testing light level not used during the training. Despite the same temporal sequence of checkerboard noise being presented at each light level, RGCs exhibited distinct responses, indicative of adaptation (Fig. 4b). The model accurately predicted responses to movies at the two training light levels (FEV of 93% and 83% for an example RGC, Fig. 4c, columns 1–2), with median FEVs of 84% ± 11% for high and 78% ± 3% for the medium light level (Fig. 5a). However, this model performed poorly at the low testing light level (Fig. 4c, column 3) with an FEV of only 24% ± 15% across cells (Fig. 5a).

Alterations in ambient light levels induce adaptation in the retina that alters both the sensitivity and kinetics of RGC responses[24,27,28]. CNN models can effectively capture linear sensitivity changes across global light levels, evident by consistent amplitudes of the predicted responses across light levels differing by a log unit (Fig. 4c). This is in part achieved by Layer Norm at the input to the model that discounts the mean intensity from input stimuli, resulting in a simple gain change commensurate with Weber adaptation. The failure of the CNN model

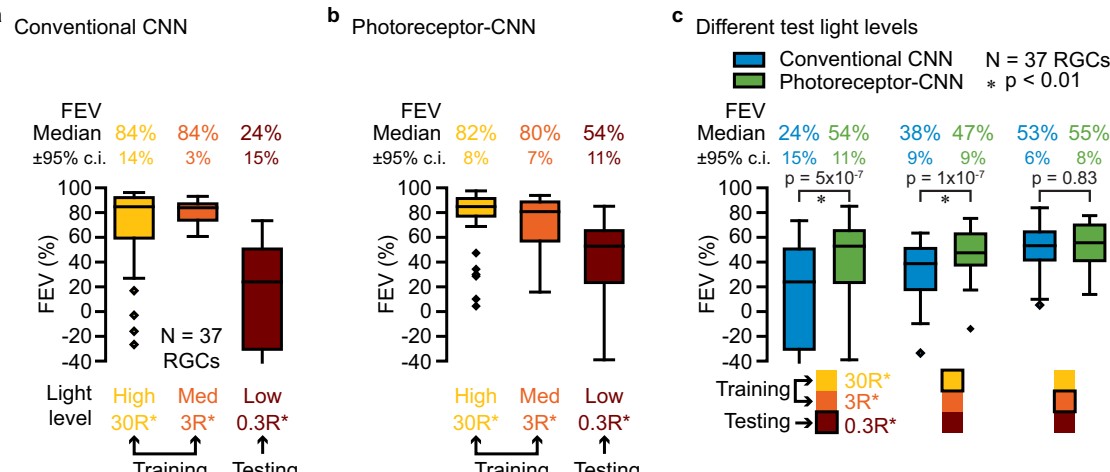

**Fig. 5 | Incorporating photoreceptor adaptation enables CNNs to generalize across light levels.** Performance of (**a**) a conventional CNN model, and (**b**) the photoreceptor–CNN model. Each model was evaluated at three light levels (labeled below each box plot): high (column 1; yellow) and medium(column 2; orange), at which the models were trained, and low (column 3; red) which the models did not see during the training. The box plots indicate the median FEV across 37 RGCs, interquartile range (25th and 75th percentiles), and minima and maxima within 1.5 times the interquartile range. Outliers are plotted as individual points. Numbers at the top of each box plot are the median FEVs ± 95%c.i. **c** Performance of the conventional CNN model (blue color; same model as in **a**), and the photoreceptor–CNN model (green color; same model as in **b**) at all combinations of training and test light levels. For each column, the legend below the box plot panel shows the two light levels the models were trained at and the third light level at which it was tested (black outline). The box plots show the distribution of FEVs at this testing light level. Testing light levels were low (column 1), high (column 2), and medium (column 3). The photoreceptor–CNN model and the conventional CNN model showed statistically significant differences when tested at low light level (column 1; $p = 5 \times 10^{-7}$) and high light level (column 1; $p = 1 \times 10^{-7}$). $p$ values were calculated by performing a paired two-sided Wilcoxon signed-rank test on the FEV distributions ($N = 37$) from the CNN and photoreceptor–CNN model at each testing light level. Source data are provided as a Source Data file.

at the low test light level (Fig. 5a, column 3) can therefore be primarily attributed to nonlinear sensitivity changes and shifts in response kinetics across the light levels, aspects not captured by Layer Norm alone. This is most prominent at the low test light level, where RGC responses slow considerably (Fig. 4b inset) and the inability of conventional CNNs to adaptively regulate both response sensitivity and kinetics limits their performance to generalize to this condition.

We next sought to determine whether incorporating the photoreceptor layer at the input stage could improve the ability of the CNNs to alter their sensitivity and kinetics in a light intensity-dependent manner and better predict the experimental data. To test this, we subjected the photoreceptor–CNN to the same test as the conventional CNN (above): we trained the model end-to-end (including the photoreceptor layer initialized to reflect rod photoreceptor parameter values) to predict primate RGC responses to the binary checkerboard movie at the high and medium light levels. Similar to the conventional CNN model, the photoreceptor–CNN model reliably predicted responses to held-out stimuli at the two training light levels (Fig. 4d columns 1–2 and Fig. 5b columns 1–2). Importantly, the photoreceptor–CNN model could also explain 54% ± 11% of the variance in responses at a light level lower than those at which it was trained (Fig. 4d column 3 and Fig. 5b). This performance was a two-fold improvement over the conventional CNN model without the photoreceptor layer ($p = 5 \times 10^{-8}$, Wilcoxon signed-rank test, $N = 37$ RGCs), which only explained 24% ± 15% of the variance (Fig. 5a). We attribute this to the model's ability to modulate output properties, like the response kinetics, based on mean light level (see below).

Although we observed improved performance at the test light level, such gains were not evident at the training light levels (Fig. 5a, b). This is unlike our previously described experiments with the naturalistic movies (Fig. 3a) where the photoreceptor–CNN outperformed the conventional CNN even at the training light level. Noise stimuli have a limited range of contrasts than natural scenes and lack temporal correlations. Hence, photoreceptor responses to noise stimuli at a single mean light level are nearly linear, with minimal changes in adaptation

state. This leads to similar performance across conventional CNN and photoreceptor–CNN models (Fig. 5a, b) at the training light levels.

We also examined the performance of the photoreceptor–CNN across all different combinations of training and testing light levels: the model was trained on data from two light levels, and then evaluated on test data from a different light level (Fig. 5c; Supplementary Fig. 2 shows the population data for ON and OFF RGCs separately). For all combinations of training and testing light levels, the photoreceptor–CNN model generalized better to new light levels than the conventional CNN model without the photoreceptor layer. The difference was smallest for the case where the testing light level (medium) was intermediate between the high and low training light levels (Fig. 5c, column 3; $p = 0.83$, two-sided Wilcoxon signed-rank test, $N = 37$ RGCs). This is unsurprising given that conventional CNN models can interpolate between different sets of training conditions. Moreover, the similarity in responses at the high and medium light levels (inset in Fig. 4b) means that in the interpolation condition, the model predictions at the testing light level need not differ much from those at one of the training light levels. However, in the more challenging extrapolation tests—and especially in extrapolation to the low light level at which the response kinetics are appreciably different—the photoreceptor–CNN performs much better because the photoreceptor layer enables the models to adjust their output in a light-level-dependent manner.

## Photoreceptor–CNN model captures light-level-dependent changes in response kinetics

Normalization layers like Layer Norm allowed both the conventional CNN and photoreceptor–CNN models to capture steady-state sensitivity changes across light levels (Fig. 4c, d). But the inability of these normalization layers to adjust the kinetics of the predicted responses suggested that the observed performance gains with the photoreceptor–CNN model resulted from its intrinsic ability to capture light-dependent changes in RGC response kinetics. For instance, measured RGC responses were faster (Fig. 6a) and temporal receptive

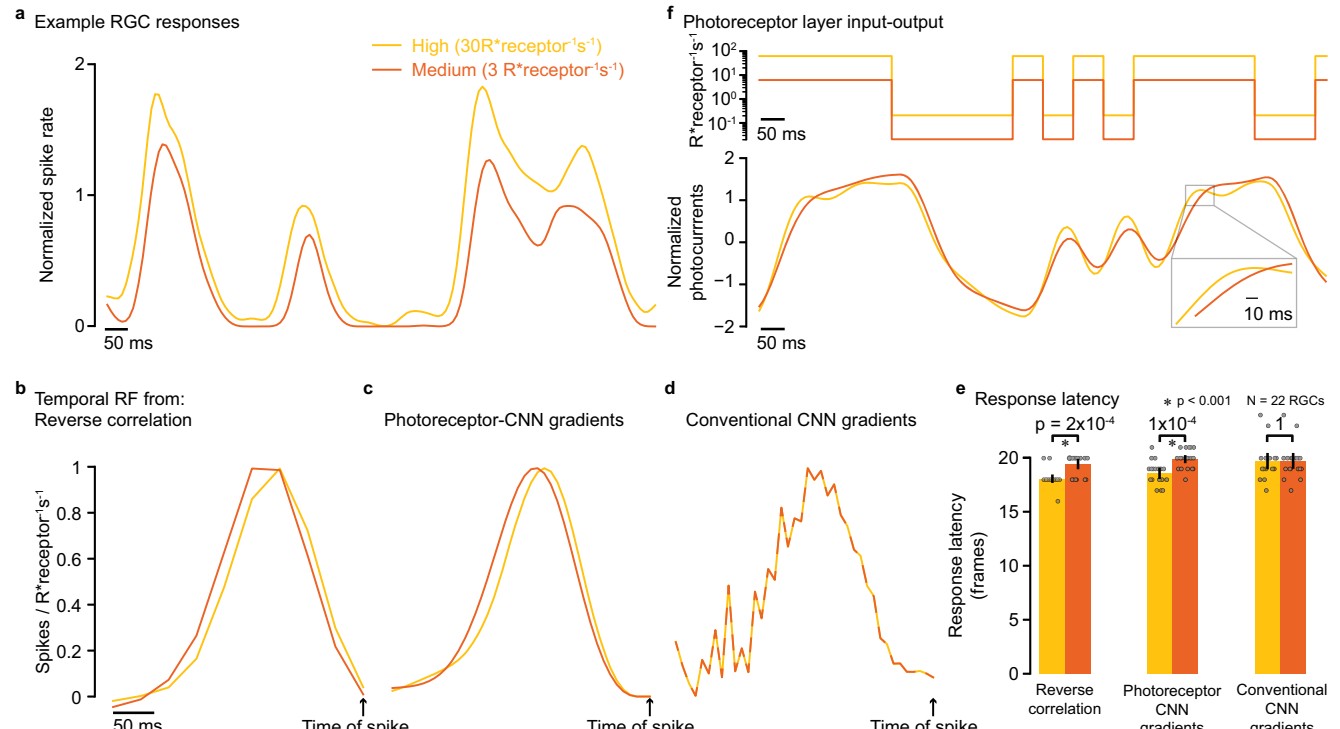

**Fig. 6 | Photoreceptor layer enables CNNs to adjust their response kinetics in a light-level-dependent manner. a** Normalized recorded spiking activity of an example RGC in response to a white noise stimulus at two light levels used in model training: high (30 R*receptor$^{-1}$s$^{-1}$; yellow), and medium (3 R*receptor$^{-1}$s$^{-1}$; orange). **b** Temporal receptive field of the same RGC calculated using reverse correlation of a white noise movie (55 min) at the two different light levels. **c** Temporal receptive fields of the same RGC obtained by averaging across multiple instantaneous temporal receptive fields from photoreceptor–CNN output gradients with respect to multiple input movie segments. **d** Same as in (**c**) but for the conventional CNN model. **e** Mean response latency ($N$ = 22 RGCs), calculated as the number of frames (1 frame = 8 ms) between the time of spike and peak of the temporal receptive field. Error bars indicate 95% confidence interval of the mean. Gray circles indicate the individual data points ($N$ = 22 RGCs). Colors (legend in **a**) represent the light level at which temporal receptive fields were calculated from reverse correlation of

experimental data (left column), photoreceptor–CNN model gradients (middle column) and conventional CNN model gradients (right column). An asterisk indicates a statistically significant difference in response latencies between two light levels ($p < 0.001$, $N$ = 22 RGCs, two-sided Wilcoxon rank-sum test). Response latency was significantly different across the two light levels when calculated from reverse correlation (column 1; $p = 2 \times 10^{-4}$) and photoreceptor–CNN model gradients (column 2; $p = 1 \times 10^{-4}$). **f** Top. Intensity changes over time for a single pixel in the binary checkerboard white noise movie at the two different mean light levels. Bottom. The output of the photoreceptor layer from the photoreceptor–CNN model after the Layer Normalization layer that immediately follows the photoreceptor layer. This output is fed into subsequent CNNs. Inset zooms the lag in photocurrents at medium light level (compare orange and yellow lines). Legend in (**a**) is valid for all panels (line style varies for clarity). Source data are provided as a Source Data file.

fields had shorter latencies at the higher light level (Fig. 6b; estimated by reverse correlation; Methods). The response latency was consistently lower for the higher light level across the population of RGCs (Fig. 6e, left column). We tested whether the trained photoreceptor–CNN and conventional CNN models captured these observed changes in response kinetics (Fig. 6b, e column 1). The temporal receptive fields of model RGCs (see Fig. 6c, d for an example RGC) were calculated by averaging the instantaneous temporal receptive fields (Supplementary Fig. 4a) across all movie segments. These instantaneous receptive fields were estimated by computing the gradients of the predicted RGC firing rates with respect to each input pixel value[12,13] and then decomposing the resulting spatiotemporal receptive fields into spatial- and temporal components (Methods). We restricted this analysis to a subset of 22 RGCs that displayed FEV values greater than 50% at the training light levels (high and medium light levels) for both models (models of Figs. 4, 5a, b). Thus, these were the RGCs for which the predictions from both models were most reliable, facilitating estimates of receptive fields from model RGC gradients with respect to input stimuli.

Consistent with our hypothesis, the temporal receptive fields of photoreceptor–CNN model RGCs exhibited intensity-dependent changes in response kinetics, with shorter latencies at the higher light level (Fig. 6c). This latency difference was statistically significant at the population level (Fig. 6e column 2; $p = 1 \times 10^{-4}$, two-sided

Wilcoxon rank-sum test, $N$ = 22 RGCs). This change in response latency could already be observed at the output of the photoreceptor layer (Fig. 6f), simply as a function of input light intensity. In contrast, the conventional CNN model showed no changes in latencies across the two light levels at which the models were trained (Fig. 6d, e column 3). This analysis underscores the effectiveness of the photoreceptor layer in capturing and adapting to dynamic changes in response kinetics associated with varying light conditions. In addition, photoreceptor–CNN models can also better capture sensitivity changes across light levels (see Supplementary Note 3).

## Incorporating photoreceptor adaptation enables generalization across photopic and scotopic light levels
Having observed that the photoreceptor–CNN model generalizes well across light levels that differ by 1–2 orders of magnitude (Fig. 5c), we wondered whether it could also generalize across more extreme variations in lighting. To answer this question, we trained conventional CNN and photoreceptor–CNN models to predict rat RGC responses to noise stimuli at a relatively bright photopic light level (10,000 R*receptor$^{-1}$s$^{-1}$) where cone photoreceptors predominantly contribute to vision, and evaluated the ability of the model to generalize to the much dimmer (scotopic) light level (1 R*receptor$^{-1}$s$^{-1}$) where rod (and not cone) photoreceptors are active (Fig. 7a). For this analysis, we used rat RGC recordings that we previously published in ref. 28.

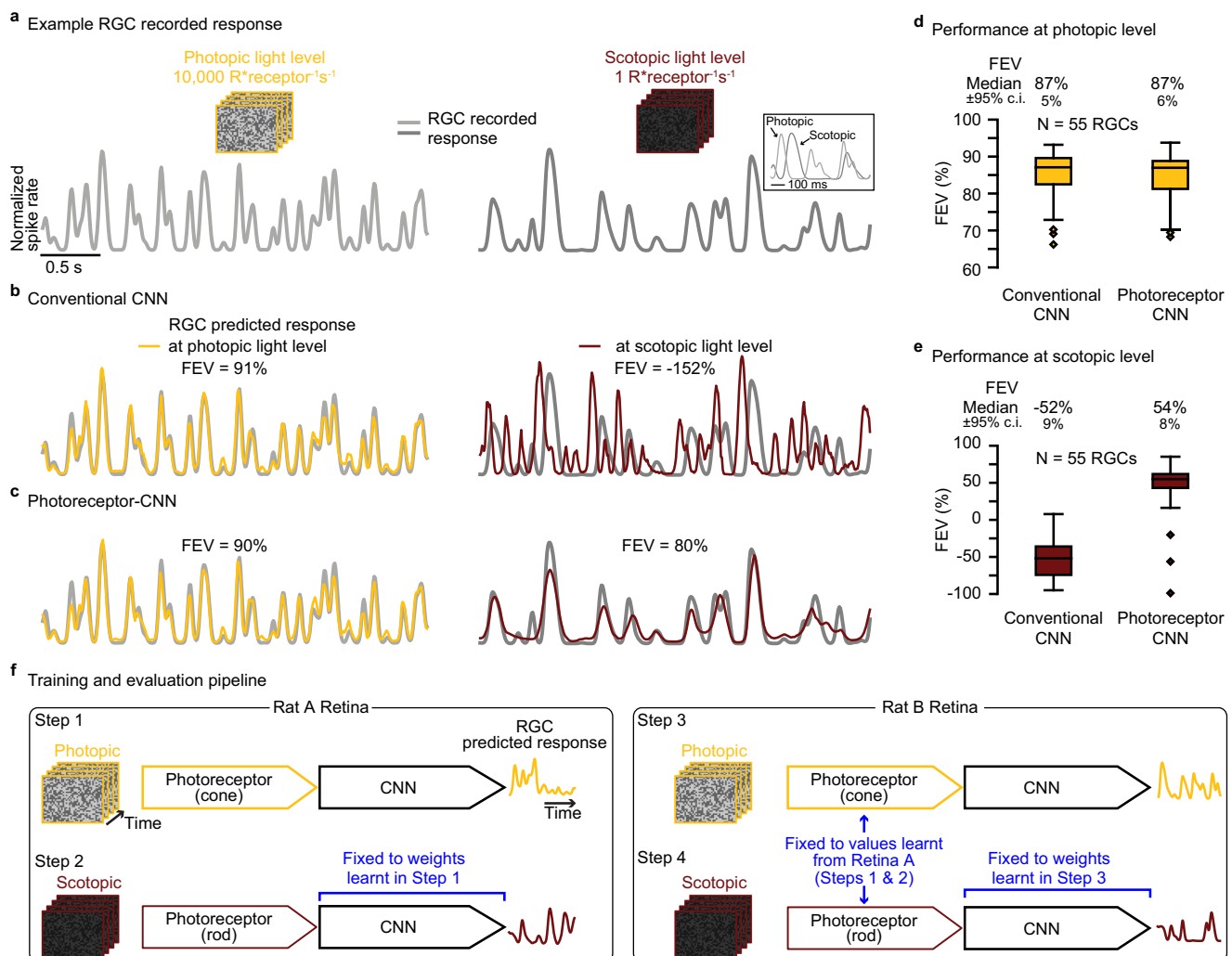

**Fig. 7 | Incorporating photoreceptor adaptation enables CNN to generalize across extremely different light levels. a** Rat Retina B example RGC recorded responses shown as normalized spike rate (gray line) to held-out white noise at photopic light level (left; yellow stimulus outline) and scotopic light level (right; red stimulus outline). Inset shows an overlay of a response segment at both light levels. **b** Responses predicted by a conventional CNN model (colored lines) at the two light levels in (**a**) when trained only at photopic light level. FEV values above each trace quantify the model's performance for this RGC. **c** Same as in (**b**) but for the proposed photoreceptor–CNN model in which the CNN layers were only trained at photopic light level. Performance of conventional CNN model (left) and the photoreceptor–CNN model (right) when trained at photopic light level and evaluated at (**d**) photopic and (**e**) scotopic light levels. The box plot indicates the median FEV across 55 RGCs (Retina B), the interquartile range (25th and 75th percentiles), and minima and maxima values, which are determined as the smallest and largest observations within 1.5 times the interquartile range. Outliers are plotted as individual points. Numbers at the top of each box plot are the median FEVs ± 95%c.i. **f** Schematic for training across extremely different light levels. **Step 1:** PR-CNN model was trained end-to-end to predict Rat Retina "A" RGC responses at photopic light level, estimating cone photoreceptor parameters and the inner retina circuit (the CNN layers). **Step 2:** The model was re-trained at scotopic light level with fixed CNN layers from Step 1, learning rod parameters. In this case, the photoreceptor model learnt parameters reflecting rods. **Step 3:** The model was trained to predict Rat Retina "B" responses at photopic light level with photoreceptor parameters fixed to cone parameters learnt in Step 1. **Step 4 (testing):** The model was tested to predict Rat Retina "B" responses at scotopic light level using rod photoreceptor parameters learnt in Step 2 and inner retina pathways from Step 3. Source data are provided as a Source Data file.

The conventional CNN model trained at the photopic light level could reliably predict RGC responses to held-out data at the photopic light level (example RGC responses in Fig. 7b, left; population data in Fig. 7d). However this model badly failed (FEV of −52% ± 9%; $N = 55$ RGCs) to predict responses at the scotopic light level (Fig. 7b, right and Fig. 7e). The proposed photoreceptor–CNN model, however, did surprisingly well (Fig. 7c, right), achieving FEV of 54% ± 8% on this task (Fig. 7e). For this experiment, we first trained the photoreceptor–CNN model at high light level and then replaced that model's photoreceptor parameters (which correspond to cone cells at this light level) with those corresponding to rod cells (as explained in Fig. 7f and Supplementary Note 4). The remaining CNN parameters were unchanged by this procedure. This finding demonstrates that the changes in

photoreceptor layer parameters alone can account for much of the difference in how the photoreceptor–CNN model predicts steady-state RGC responses at these two light levels.

## Discussion
We introduced a new CNN layer for vision models that build upon a biophysical model of phototransduction[20]. When used as a front-end to CNNs, this photoreceptor layer allows the CNN outputs to adapt to the prevailing inputs in a manner that more accurately mimics the retina. Consequently, the photoreceptor–CNN models surpass conventional CNN models at predicting RGC responses to naturalistic movies that simulate rapid local changes in light intensity due to eye movements, and at predicting responses across steady-state changes

in mean light levels. The improved performance could not be replicated by replacing the biophysical photoreceptor model with a linearized photoreceptor model. Thus, the success of the biophysical photoreceptor–CNN model is attributable to nonlinear processes governing adaptation within the biophysical photoreceptor model.

ANNs, of which CNNs are a sub-class, are universal function approximators[29] and therefore in principle they are capable of implementing any transformation with simple nonlinear units. This suggests that, in principle, a sufficiently large ANN can accurately model neural responses to stimuli with the same statistics as the training set, without the need for any bio-inspired adaptive mechanisms. Nonetheless, ANNs can benefit from having the right inductive biases that represent prior knowledge about the underlying data, as demonstrated by the benefits of CNNs in computer vision over non-convolutional forms of ANN[30–32]. In the same way, our results demonstrate that equipping CNN-based deep learning models with photoreceptor adaptive mechanisms improves their ability to capture retinal responses to stimuli with local luminance fluctuations, and enables them to better generalize to out-of-distribution tasks, such as extrapolating to new lighting conditions (Fig. 5c columns 1, 2, Fig. 7e). The new photoreceptor–CNN is much better at this challenging task as the photoreceptor layer enables the CNN to learn response properties such as the dependence of kinetics on light intensity (refs. 20–22,33; Fig. 6c). This capability is demonstrated by the difference in kinetics of the responses at 3 R*receptor$^{-1}$s$^{-1}$ and 30 R*receptor$^{-1}$s$^{-1}$ (Fig. 6f, bottom); a fully linear model predicts identical kinetics at different mean light levels.

In addition to improving generalization between light levels, the photoreceptor–CNN model also substantially improves performance for predicting responses to high-resolution naturalistic stimuli at a single light level (Figs. 2, 3). However, it still falls short of perfectly predicting retinal output. This is apparent in its inability to match the performance achieved in experiments where the models were trained and evaluated using white noise movies only (Fig. 5b, columns 1, 2). The lower performance with naturalistic movies might be due to the disparity in training data (8 min of naturalistic movies in experiments of Fig. 3 versus 40 min of white noise movies in experiments of Fig. 5), but other factors could also contribute. For example the model lacks retinal adaptive mechanisms found downstream of the photoreceptors, such as spike frequency adaptation in RGCs[34,35], which may be required to capture the wider range of contrasts and temporal correlations present in naturalistic movies but absent in white noise stimuli. Lack of such downstream adaptive mechanisms may also explain why the photoreceptor–CNN model could not achieve the same level of performance at held-out test light levels as it did at the light levels used for training (Fig. 5b). Introducing adaptive recurrent units (ARU; developed by ref. 36) to the output layer of the photoreceptor–CNN, which implement spike frequency adaptation through dynamic control of a nonlinearity, is a potential solution. ARUs at the output layer would also enable the network to have output units with diverse properties, similar to the diversity[37,38] of RGCs.

Our current model also does not explicitly capture the intricacies of adaptation in the intervening circuitry between photoreceptors and RGCs (i.e., in bipolar and amacrine cells). These include changes in gap junction coupling and switching between linear and nonlinear spatial summation (i.e., subunit rectification[39–41]). To capture this adaptation, an adaptive-convolution layer based on a model for divisive gain control[22,42] could be used. This trainable layer would incorporate two pathways with distinct kinetics, with the output of one pathway controlling the sensitivity of the other, allowing for greater adaptability to changes in stimuli.

Another limitation of the current approach is that Layer Norm was retained at the inputs of CNN in the photoreceptor–CNN model, normalizing the photoreceptor output. While this may inadvertently mitigate sensitivity changes across light levels, typically managed by

the photoreceptor layer, it serves a crucial role in compensating for disparities in scale between the parameters of the biophysical model and the downstream CNN weights. The absence of Layer Norm negatively impacts the convergence of the photoreceptor–CNN model. Here, Layer Norm can, in principle, account for linear sensitivity changes linked to the mean intensity of a training sample (640 ms movie), but will have minimal impact on mitigating dynamic sensitivity changes triggered by fluctuations in pixel intensities within a training sample.

ANNs offer the potential to simulate networks of biological neurons, including those in the retina or visual cortex, making them highly relevant for visual neuroscience. These models are capable of automatically learning meaningful representations through multiple layers of abstraction. Establishing correspondence between ANN layers and neural layers[8,9,13,43,44] is increasingly providing insight into biological circuits. One challenge in using current ANNs to elucidate biological circuits is that ANNs are primarily designed to optimize performance on specific tasks rather than to mimic biological circuits. As a result, the structure and function of ANNs may not accurately reflect the complexity and organization of biological circuits which often include feedback loops and dynamic interactions between neurons. Moreover, ANNs typically consist of homogeneous units repeated throughout the network, which oversimplifies real neurons and neural circuitry and may fail to capture their full complexity. In contrast, the biophysical phototransduction model we use in the photoreceptor layer has parameters that map directly onto the biology, providing an opportunity for investigating the role of photoreceptor adaptation in the retina. For example, slower rod-mediated RGC responses at dim conditions compared to cone-mediated responses at brighter light levels[28,45] may explain the temporal lag between predicted and actual response at dim light level (Fig. 7b, right): the conventional CNN model trained at bright light levels (10,000 R*receptor$^{-1}$s$^{-1}$) learned the faster kinetics of the cone pathway. While some of the differences in RGC responses may arise due to faster cone response kinetics[46–49], the relative contribution of photoreceptors and downstream retinal adaptation are not well understood. Fixing the photoreceptor layer parameters to empirically measured values, (like in Fig. 3c), can help in distinguishing photoreceptor from circuit mechanisms. Similarly, such biologically plausible models could provide insights into mechanisms underlying neural adaptation in other areas of the brain.

While our current findings indicate that the photoreceptor model we used offers superior performance (Fig. 3a) vs simpler photoreceptor models (like the linearized model shown in Fig. 3b), we acknowledge that other empirical models capturing similar dynamics may perform similarly well on the retinal prediction task. It is also possible that recurrent artificial units like the long short-term memory may partially capture photoreceptor adaptation effects by keeping track of arbitrary long-term dependencies in the input sequences. However, these units demand significantly more training data and introduce tens of thousands of parameters to the neural network. In stark contrast, the proposed biophysical photoreceptor layer only adds 12 parameters. Further, these parameters could be fixed based on direct photoreceptor recordings with minimal changes in CNN performance (Fig. 3c). In addition, the overarching goal is to integrate neural biophysics into ANNs to develop biologically interpretable computational models that surpass the limitations of conventional ANNs, offering a comprehensive framework for understanding complex biological phenomena.

In a similar vein, other neural predictor architectures, such as Generalized Linear Models could be used instead of CNNs, in conjunction with the photoreceptor model. Notably, we do not consider the CNN stages to be essential to our approach: rather, we consider the CNN to be the flexible scaffolding for incorporating the photoreceptor model into a trainable retina model. In future, this will allow incorporating fully-trainable biophysics models of downstream retinal

components into this scaffolding. The result will be models with biologically-interpretable components that can predict retinal responses with high accuracy under varied conditions. We anticipate that these models could have substantial benefits for mechanistic investigations of visual function.

In general, models of retina that can leverage deep learning to model multiple ganglion cells simultaneously, together with biologically interpretable components, could be used to dissect the relative contributions by different cell types in the retina. They could also serve as an input stage to visual-cortical models to investigate higher visual processing under dynamic conditions. From a wider neuroscience perspective, this approach demonstrates the power of integrating neural dynamics in ANNs modeling brain functions where biophysical layers match the sensitivity to changing input conditions, while the downstream layers extract relevant features from dynamically adapting input stages. In summary, this approach establishes a framework to test which biological components are required to replicate brain function. Beyond neuroscience, these models could also pave the way for medical interventions, such as prosthetic devices that restore sight to the blind[50].

## Methods

### Retina electrophysiology

Retina electrophysiology experiments were performed in two different labs. In the Manookin Lab, electrophysiological experiments were performed using ex vivo retina from a 11 years old female macaque (*Macaca nemestrina*) obtained through the tissue distribution program at the University of Washington National Primate Research Center and in accordance with the Institutional Animal Care and Use Committee at the University of Washington. Additional primate (17 years old male *Macaca mulatta*) and rat (2 female Long-Evans) retina electrophysiology experiments were performed in the Field Lab at Duke University, in accordance with Duke University's Institutional Animal Care and Use Committee.

Electrophysiology experiments followed similar procedures in both the labs. For primate electrophyiology experiments, eyes were enucleated from a terminally anesthetized macaque monkey and hemisected, and the vitreous humor was removed. Immediately after enucleation, the anterior portion of the eye and the vitreous were removed in room light. The eye cup was placed in a dark sealed container with Ames' solution (Sigma, St. Louis, MO) at room temperature. Under infrared illumination, segments of peripheral retina 6–15 mm (25–70°, 200 μm/°) from the fovea and 3–5 mm in diameter were dissected and isolated from the retinal pigment epithelium. Preparation for the rat retinae was similar and described in detail in ref. 28. Briefly, the retina of an euthanized animal was extracted and dissections were performed in darkness with the assistance of infrared converters. We dissected dorsal pieces of the retina that were $3 \times 2$ mm large. For recording, the retina was kept at 32–35 °C and was perfused with Ames' solution bubbled with 95% $O_2$ and 5% $CO_2$, pH 7.4.

The segment of retina was then placed flat, RGC layer down, on a planar multielectrode array (MEA) covering an area 2000 μm × 1000 μm. The MEA consisted of 512 electrodes with 60 μm or 30 μm spacing. Spikes on each electrode were identified by thresholding the voltage traces at 4 s.d. of a robust-estimate of the voltage s.d. For retina experiment involving naturalistic stimuli (Manookin Lab), spike

sorting was performed using the Kilosort[51] software package (version 2.5). Spike waveform clusters were identified as neurons only if they exhibited a refractory period (1.5 ms) with < 1% estimated contamination. For retina experiments across light levels (Field Lab), spike sorting was performed by an automated PCA algorithm and verified by hand with a custom software[52,53]. Spike waveform clusters were identified as neurons only if they exhibited a refractory period (1.5 ms) with < 10% estimated contamination. Most sorted units from the primate retina had 0% spike contamination based on refractory period violations. Other units had contamination in the range of 0.05–0.09% with one unit at 0.8%. Only units that could be reliably tracked across all recording conditions were considered for further analysis.

For each retinal segment, a Retinal Reliability Index was computed to assess tissue quality. This involved analyzing the responses of individual sorted RGCs to multiple trials using either white noise or naturalistic movies. We first estimated the trial-averaged noise by categorizing trials into two groups, averaging responses within each subgroup, and determining the mean squared error as

$$\sigma_{noise}^2 = E_t[(y_t^A - y_t^B)^2] \tag{1}$$

where, $y_t^A$ and $y_t^B$ are the observed spike rates of an RGC calculated as an average across set of trials $A$ and set of trials $B$ respectively at time bin $t$. The sets $A$ and $B$ were obtained by randomly splitting the total number of repeats into two. We then computed the fraction of explainable variance which is the fraction of variance of each RGC attributable to the stimulus as

$$\text{Fraction Explainable Variance} = \frac{Var[y^A] - \sigma_{noise}^2}{Var[y^A]} \tag{2}$$

where,

$$Var[y^A] = \frac{1}{T}\sum_{t=1}^{T}(y_t^A - \bar{y}^A)^2 \tag{3}$$

and $y^A$ is the observed spike rate at time bin $t$ and $\bar{y}^A$ is the mean firing rate across time. A higher fraction indicates recordings with low noise, where most of the variance in RGC responses is stimulus-driven. The retinal reliability index was determined by computing the median of the fraction of explainable variance across all sorted RGCs in each experiment. These values are presented in Table 1 for each retina used in this study. Intuitively, higher the value (maximum 1), better the quality of recordings.

### Visual stimulation and data acquisition for primate retina experiment using naturalistic movies (Figs. 2, 3)

Visual stimuli were created with custom Matlab code. Stimuli were presented with a gamma-corrected OLED display (Emagin, Santa Clara, CA) refreshing at 60.32 Hz. The display had a resolution of 800 × 600 pixels covering 3.0 × 2.3 mm on the retinal surface.

Spectral intensity profile (in μWcm⁻² nm⁻¹) of the light stimuli was measured with a calibrated CCS100 spectrometer (Thorlabs). We transformed the stimulus intensity into equivalents of photoisomerizations per receptor per second (R*receptor⁻¹s⁻¹). The spectrum was converted to photons cm⁻² s⁻¹ nm⁻¹, convolved with the

## Table 1 | Retinal reliability index for each retina used in this study

| Retina | Retinal Reliability Index | Sorted RGCs |
|---|---|---|
| Macaque retina 1: Natural stimuli experiments (Figs. 2, 3) | 0.90 | 57 |
| Macaque retina 2: Across light level experiments (Figs. 4, 5) | 0.96 | 37 |
| Rat retina 1: Across light level experiments (Fig. 7) | 0.99 | 61 |
| Rat retina 2: Across light level experiments (Fig. 7) | 0.99 | 58 |

normalized spectrum of macaque cones and rods, and multiplied with the effective collection area of these photoreceptors. The ambient light level (i.e., mean stimulus intensity) was set using neutral density filters in the light path. The attenuation of each neutral density filter was measured for the red, green, and blue LEDs using a calibrated UDT 268R radiometric sensor (Gamma Scientific).

We recorded RGC activity to 36-min of binary checkerboard white noise stimuli and 9-min of gray scale naturalistic movies at mean light level of 50 R*receptor$^{-1}$s$^{-1}$. The checkerboard stimuli in this experiment had 100 × 75 pixels, where each pixel edge corresponded to 30 μm on the retina surface. The refresh rate of the stimulus was set to 60.32 Hz (~16.6 ms per frame). The naturalistic movies were created by displacing natural scene images from Van Hateren dataset[54] across the retina, incorporating eye movement trajectories derived from the DOVES dataset[55] (Fig. 1a). We used nine different natural scene images leading to nine naturalistic movies. Each movie was 6-s long where the image remained stationary for the first 1-s to allow time to adapt to the spatial contrast before the motion began, which lasted for 5-s. The nine movies were played in sequence and the entire sequence was repeated ten times, totaling a duration of 9 min for naturalistic movies. These movies were presented to the retina at a resolution of 800 × 600 pixels where each pixel edge corresponded to ~3.8 μm on the retina surface. We selected 57 RGCs (27 ON and 30 OFF parasol cells) for modeling purposes based on spike sorting quality and reliability across experimental conditions.

At the model training stage, each repeat of the movie was treated as an individual movie, i.e., 9 movies with 10 trials were treated as 90 movies. 80 of which (8 unique movies and 10 trials) were used for training the model and the held-out movie was used to validate the model against trial averaged responses. Additionally, naturalistic movies were spatially down-sampled by a factor of 8 to 100 × 75 pixels to match the resolution of checkerboard white noise stimuli. This was necessary as we first trained the models on checkerboard movie and then fine-tuned the same model with naturalistic movies.

### Visual stimulation and data acquisition for primate and rat retina experiment at different light levels (Figs. 4–7)

Visual stimuli were created with custom Matlab code. Stimuli were presented with a gamma-corrected OLED display (SVGA + XL Rev3, Emagin, Santa Clara, CA) refreshing at 60.35 Hz. The image from the display was focused onto the photoreceptors using an inverted microscope (Ti-E, Nikon Instruments) with a ×4 objective (CFI Super Fluor ×4, Nikon Instruments). The optimal focus was confirmed by presenting a high spatial resolution checkerboard noise stimulus (20 × 20 μm, refreshing at 15 Hz) and adjusting the focus to maximize the spike rate of RGCs over the MEA. The display had a resolution of 800 × 600 pixels covering 4 × 3 mm on the retinal surface.

Spectral intensity profile (in μW cm$^{-2}$ nm$^{-1}$) of the light stimuli was measured with a calibrated Thorlabs spectrophotometer (CCS100). We transformed the stimulus intensity into equivalents R*receptor$^{-1}$s$^{-1}$ by converting the power and the emission spectra of the display to an equivalent photon flux by Planck's equation. This converted the emission spectrum to photons cm$^{-2}$ s$^{-1}$ nm$^{-1}$, which was then convolved with the normalized spectral sensitive of rods[56], and multiplied with the effective collection area of rods (0.5 μm$^2$). The ambient light level (i.e., mean stimulus intensity) was set using neutral density filters in the light path. In each recording, stimuli were first presented at the darker light level. For every subsequent higher light level, the retina tissue was first adapted to that light level before continuing the recordings.

Stimuli consisted of non-repeated, binary checkerboard white noise interleaved with repeated ($N = 126$ or 225 trials), binary white noise segments (5 or 10 s) to estimate noise. The total duration of stimulation was 60 min. We recorded primate RGC activity to the same 60 min white noise sequence at three different mean light levels, each

differing by 1 log unit: 0.3 R*receptor$^{-1}$s$^{-1}$, 3 R*receptor$^{-1}$s$^{-1}$ and 30 R*receptor$^{-1}$s$^{-1}$. These light levels fall under the scotopic regime, where mostly rod photoreceptors contribute to vision. The movies across the three light levels only differed in their mean pixel values which were 0.3 R*receptor$^{-1}$s$^{-1}$ (low light level), 3 R*receptor$^{-1}$s$^{-1}$ (medium) and 30 R*receptor$^{-1}$s$^{-1}$ (high). The checkerboard stimuli in this experiment had 39 pixels × 30 pixels, where each pixel edge corresponded to ~140 μm on the retina surface. The refresh rate of the stimulus was set to 15 Hz which means that each checkerboard pattern was exposed on to the retina for ~67 ms. In this work, we used a subset of 37 recorded RGCs that could be reliably tracked across light levels and were classified as high quality units after spike sorting. This subset contained 2 ON parasol, 28 ON midget, 5 OFF parasol and 2 OFF midget RGC types).

The rat experiments of ref. 28 were performed at two light levels differing by 4 log units: 1 R*receptor$^{-1}$s$^{-1}$ (scotopic light level where mostly rod photoreceptors contribute to vision) and 10,000 R*receptor$^{-1}$s$^{-1}$ (photopic light level where cone photoreceptors predominantly contribute to vision). The white noise checkerboard movie in these experiments had 10 pixels × 11 pixels, with each pixel edge corresponding to ~252 μm on the retina. The refresh rate of the stimulus was 60 Hz and 30 Hz at the photopic and scotopic light levels, respectively. In this work, we used data from two rat experiments: a subset of 61 RGCs from Retina A and a subset of 55 RGCs from Retina B that could be reliably tracked across light levels and were classified as high quality units after spike sorting. This subset contained OFF brisk sustained and OFF brisk transient RGC subtypes.

### Data preprocessing for models

Both white noise and naturalistic movies were up-sampled to 120 Hz by repeating each frame so that each frame had a duration of 8 ms. This up-sampling was necessary for the photoreceptor layer in which differential equations are solved using the Euler method.

Spikes were grouped in 8 ms time bins spanning the duration of the movie. Firing rates were then estimated by convolving the binned spike counts with a Gaussian of $\sigma = 32$ ms (4 frames/bins) standard deviation and amplitude of $0.25\sigma^{-1}e^{1/2}$. The resulting firing rates for each RGC were normalized by the median firing rate of that RGC over the course of the experiment. This was done to ensure that responses of all output units of the model (i.e., the modeled RGCs) were at the same scale.

### Conventional CNN architecture

The general architecture of the conventional CNN we used was similar to Deep Retina[9]. The model (Fig. 1b) had three convolution layers (orange color), followed by a fully-connected output layer (black arrows). The model takes as input a movie (80 frames per training example where each frame corresponds to 8 ms) and outputs an instantaneous spike rate for each RGC at the end of that movie segment. The first convolution layer is a 3D convolutional layer operating in both the spatial and temporal dimensions. The output of the 3D convolutional layer is a 2D image which is normalized using Batch Normalization (BatchNorm) and then passed through a rectifying nonlinearity. All the temporal information from the movie is extracted by this layer as the temporal dimension of the convolutional filter is the same as the temporal dimension of the movie. To down sample the spatial dimensions, we applied a 2D max pool operation (blue color) that took the maximum value over 2 × 2 patches of the previous layer's output. The subsequent 2D CNN layers are followed by a final, fully connected layer with softplus activation function that outputs the predicted spike rate for each RGC in the dataset.

To obtain the time series of RGC responses to longer movie stimuli, we feed into the model many 80-frame video samples taken from that longer movie, that correspond to 1-frame shifts. I.e., the model receives as inputs frames 1–80, 2–81, 3–82, etc., and outputs RGC responses at the times of movie frames 80, 81, 82, etc.

A Layer Normalization (Layer Norm) at the input of the first convolutional layer was applied to z-score each frame of a movie segment. Layer Norm computes normalization statistics for each pixel over its temporal history within a single training example i.e., a single movie segment comprising 80 frames. This step removes the mean luminance from each training example, mitigating sensitivity changes associated with global luminance changes while preserving the temporal structure within each movie segment.

Each convolution operation is followed by Batch Normalization which contributes to stable training and faster convergence of the model[57]. During model training, the distribution of inputs to a layer undergoes changes as the network's parameters are updated, leading to what is known as the internal covariate shift—a phenomenon that hampers model convergence and introduces instabilities. These Batch Norm layers address the internal covariate shift during training by z-scoring the input, using normalization statistics computed based on batch statistics from batches comprising over 100 movie segments. This process enforces a 0 mean and unit variance for the data, introducing two trainable parameters—shift and scale—that systematically adjust weights and biases in the CNN layer. Additionally, the running average and variance of the training data serve as non-trainable parameters saved for later use during the test phase. This normalization process mitigates extreme parameter values, preventing issues such as exploding or vanishing gradients. The scale and shift parameters enable the model to adapt to variations in feature magnitudes across layers and activations, facilitating improved and faster convergence. During the test phase, Batch Norm uses the non-trainable moving average and variance saved during the training to normalize its inputs i.e., the outputs of the convolutional layer. Batch Norm then scales and shifts the normalized input using the scale and shift parameters learned during the training phase. Notably, in the current setup, Batch Norm parameters are not influenced by the mean light level as Layer Norm at the model's input removes mean luminance from each training sample.

For modeling RGC responses across light levels (experiments of Figs. 4–7), the input to the model was a movie segment of 120 frames instead of 80 frames. The longer movie segment allowed for longer integration times at the lowest light level of 0.3 R*receptor$^{-1}$s$^{-1}$.

## Biophysical photoreceptor–CNN architecture

The proposed photoreceptor convolution layer builds upon a biophysical model of the phototransduction cascade by ref. [20] (Fig. 1c). The model incorporates the various feedforward and feedback molecular processes that convert photons into electrical signals, and therefore faithfully captures the photoreceptors' adaptation mechanisms. The biophysical model is reproduced in Supplementary Note 2 and described in brief below.

The biophysical model was represented by a set of six differential equations that mimics the enzymatic reactions of the phototransduction cascade. Rapid adaptation in this model emerges from changes in the rate of cGMP turnover produced by light intensity-dependent changes in phosphodiesterase activity and by calcium feedback to the rate of cGMP production. The model is governed by twelve parameters. By setting the model's parameter values to match experimentally-derived values from cone or rod photoreceptors, the model can be configured to represent either photoreceptor type. For all primate retina modeling in this manuscript, we configured the photoreceptor model to represent primate rods by setting the initial values of the model parameters to those that were derived from separate patch-clamp experiments[23] on primate rod photoreceptors (Supplementary Table 2). For rat retina modeling, we configured the photoreceptor model as a cone photoreceptor for modeling responses at photopic light levels, and as a rod photoreceptor for modeling responses at scotopic light levels. The parameter values here were obtained from fitting the

model to mouse cone and rod photoreceptors as part of patch-clamp experiments for other studies[23]. The corresponding values are stated in Supplementary Table 2.

We implemented this biophysical model as a fully-trainable neural network layer, called the photoreceptor layer, using the Keras[58] package in Python. All twelve parameters of the photoreceptor layer can be trained through backpropagation using the Keras and TensorFlow package in Python—although the user can also set some or all of these parameters to be non-trainable and hence held fixed at their initial value. Photoreceptor parameters were initialized to their known values (Supplementary Table 2). For the experiments presented herein, 7 of the parameters were set to be non-trainable. Some of these parameters like the concentration of cyclic guanosine monophosphate (cGMP) in darkness vary across rod and cone photoreceptor types (rods and cones). Other parameters governing cGMP conversion into current, calcium concentration in the dark, affinity for Ca$^{2+}$, and hills coefficient are comparable across photoreceptor types. The remaining five parameters, set to be trainable or non-trainable depending on the model configuration, consisted of the photopigment decay rate $\sigma$, the phosphodiesterase (PDE) activation rate $\eta$, the PDE decay rate $\phi$, the rate of Ca$^{2+}$ extrusion $\beta$ and $\gamma$ that controls the overall sensitivity of the model to light inputs. These trainable parameters also differ across photoreceptor types. In the current version of the model, the photoreceptor parameters are shared by all the input pixels, and each pixel acts as an independent photoreceptor: I.e., the conversion of each pixel into photocurrents only depends on that pixel's previous values and not on the values of the other pixels.

The photoreceptor layer converts each pixel of the input movie in units of receptor activations per photoreceptor per second (R*receptor$^{-1}$s$^{-1}$) into photocurrents (pA) by solving the differential equations using the Euler's method. Similar to the conventional CNN model, the photoreceptor layer takes as input 80 frames, where each frame corresponds to 8 ms. The output of this layer is a movie that is 80 frames long, and the same spatial dimensions as the input visual stimuli. The first 20 frames of the photoreceptor layer output are truncated to account for edge effects. The photocurrents movie is then z-scored using Layer Norm. This normalization step is crucial due to substantial differences in scale between the biophysical model's parameters and the downstream CNN weights. The absence of these normalization layers hinders the photoreceptor–CNN model's convergence. The resulting movie is then passed through the downstream CNN layers, where the size of the first convolution layer filter representing the temporal dimension is 60 frames instead of the 80 frames in the case of conventional CNN model.

For modeling RGC responses across light levels (experiments of Figs. 4–7), the input to the photoreceptor layer was a movie segment of 180 frames. The first 60 frames were then discarded to account for edge effects. The longer movie segment allowed for longer integration times at the lowest light level of 0.3 R*receptor$^{-1}$s$^{-1}$.

## Linear photoreceptor–CNN architecture

The linear photoreceptor model consists of a linear convolutional filter given by Eq. (4), and described previously in ref. [20].

$$f(t) = \alpha \left( \frac{\left(\frac{t}{\tau_{rise}}\right)^4}{1 + \left(\frac{t}{\tau_{rise}}\right)^4} \right) \times e^{-\left(\frac{t}{\tau_{decay}}\right)} \times \cos\left(\frac{2\pi t}{\tau_{osc}} + \omega\right) \quad (4)$$

The parameters for this model were initialized to the following values: $\alpha = 631$ pA/R*/s, $\tau_{rise} = 28.1$ ms, $\tau_{decay} = 24.3$ ms, $\tau_{osc} = 2 \times 10^3$ s, and $\omega = 89.97°$. These values corresponded to estimates of the single-photon response, obtained by recording cone photoreceptor responses[20]. However, all the parameters were set to trainable and could therefore be learned along with the downstream CNN weights.

Similar to the other models, the hyperparameters of the linear photoreceptor–CNN model were optimized via a grid search.

## Model training

Model weights were optimized using Adam[59], where the loss function was given by the negative log-likelihood under Poisson spike generation. The network layers were regularized with a $L_2$ weight penalty at each layer, to prevent loss of information and be more robust to outliers. In addition, a $L_1$ penalty was applied to the output of the fully-connected layer because the neural activity itself is relatively sparse and $L_1$ penalties are known to induce sparsity. Learning rates were initially set to 0.001. A learning rate scheduler reduced the learning rate by a factor of 10 at epoch 3, 30 and 100.

The number of channels in each CNN layer and the filter sizes were optimized by a grid search for each model type and dataset. Grid search for each experiment and model type was conducted using the full training data for that experiment. During the grid search procedure, models were trained for 50 epochs. Optimal hyperparameters were selected by evaluating the model on validation data that was neither used during the training phase, or during the model evaluations of predicted responses. Models with these optimal hyperparameters were then re-trained for at least 100 epochs. Optimal hyperparameters for each model used in this study are described in Supplementary Table 1.

## Model evaluation

Trained models were evaluated using the held out test dataset not seen during the training. We quantified the model performance with the fraction of explainable variance in each RGC's response that was explained by the model (FEV). This quantity (Eq. (5)) was calculated as the ratio between the variance accounted for by the model and the *explainable* variance (denominator in Eq. (5)). Such metrics to quantify how well a model predicts neural data have been used in previous studies like ref. 6. We calculate FEV as

$$FEV = 1 - \frac{\frac{1}{T}\sum_{t=1}^{T}(y_t^A - \hat{y}_t)^2 - \sigma_{noise}^2}{Var[y^A] - \sigma_{noise}^2} \quad (5)$$

where,

$$\sigma_{noise}^2 = E_t[(y_t^A - y_t^B)^2] \quad (6)$$

$y^A$ and $y^B$ are the observed spike rate of an RGC calculated as an average across a set of repeats $A$ and set of repeats $B$ respectively. The sets $A$ and $B$ were obtained by randomly splitting the total number of repeats into two. $\hat{y}_t$ represents the predicted spike rate by the model at time bin $t$. The explainable variance (denominator in Eq. (5)) is the variance of each RGC attributable to the stimulus, computed by subtracting an estimate of the observed noise from the variance across time (Eq. (7)) in the actual RGC's responses, calculated as

$$Var[y^A] = \frac{1}{T}\sum_{t=1}^{T}(y_t^A - \bar{y}^A)^2 \quad (7)$$

where $\bar{y}^A$ is the the observed spike rate $y^A$ averaged across time. In all neural data sets we considered, the number of trials was sufficient ($N = 10$ for naturalistic movies, $N = 225$ for white noise movies) and hence the estimated noise variance was quite low. As a result, our FEV values are quite similar to what is obtained using the usual fraction explained variance calculation, which does not correct for unexplainable noise. By definition, FEV can be negative if the prediction error is larger than the variance in the actual responses. We report each model's performance across all RGCs as the median FEV across the set of RGCs. For ease in interpretation, we present FEV as a percentage throughout our results.

## Model RGC temporal receptive fields (Fig. 6)

For a given model RGC, we computed the gradient of its output spiking rate with respect to the pixel values in the input movie segments, similar to ref. 13 and ref. 12. These gradients were evaluated for different binary white noise movie segments from the primate retina experiment across light levels (Figs. 4, 5). In total we had 400,000 input movie segments, generated by incrementing the white noise movie that was shown to the retina forward by one frame at a time (where 1 frame corresponds to 8 ms). These input movie segments spanned a total duration of 54 min. Since all the models were implemented with TensorFlow[60], we calculated the gradients using automatic differentiation.

The resulting gradient matrix representing spatio-temporal receptive field were decomposed into spatial and temporal components (Supplementary Fig. 3) using Singular Value Decomposition (SVD), similar to the way the spatial and temporal receptive fields are computed from the spike-triggered average (STA) analysis applied directly to experimental data.

We normalized the spatial component to have unit mean. By doing so, our process of decomposing the instantaneous spatio-temporal RF into spatial and temporal components assigned any variations in the receptive field's amplitude only to the temporal component. The average of all the instantaneous temporal receptive fields was taken as the model RGC's temporal receptive field. In Fig. 6c, d, the temporal receptive field was normalized by the maximum peak.

## Statistics and reproducibility

We used the SciPy package in Python to perform a two-sided Wilcoxon signed-rank test to compare performance across models (Figs. 3, 5). This non-parametric test was chosen due to its appropriateness for paired samples (in this case the same RGCs being modeled by different architectures) and its robustness against potential violations of normality assumptions. The null hypothesis tested was that the difference between the median fraction of explainable variance explained (FEV) by the two models for the population of RGCs modeled was 0. In comparing response latencies between two different light levels obtained by different methods (Fig. 6e), we performed a two-sided Wilcoxon rank sum test of the null hypothesis that there was no difference between the distributions ($N = 22$ RGCs) of latencies at the two different light levels.

## Reporting summary

Further information on research design is available in the Nature Portfolio Reporting Summary linked to this article.

## Data availability

The retina electrophysiology data used in this study are available under restricted access as they are part of ongoing investigations. Access can be obtained by contacting the corresponding authors. Processed data underlying each figure are provided in the Source Data file. Source data are provided in this paper.

## Code availability

Codes for the proposed photoreceptor–CNN model and for the conventional CNN model used in this study are available in the public repository https://github.com/saadidrees/dynret with identifier doi: 10.5281/zenodo.11406087 (ref. 61).

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

## Acknowledgements
We thank the reviewers for providing constructive feedback on the manuscript. This work was supported by VISTA: Vision Science to application fellowship to S.I.; Canada Research Chair grant and NSERC Discovery grant (RGPIN-2019-06379) to J.Z.; grants from the NIH (NEI R01-EY027323 to M.B.M.; NEI R01-EY029247 to E.J. Chichilnisky, F.R., and M.B.M.; NEI R01-EY028542 to F.R., NEI R01-EY031396 to G.D.F.), Research to Prevent Blindness Unrestricted Grant (to the University of Washington Department of Ophthalmology and the University of California, Los Angeles, Department of Ophthalmology).

## Author contributions
All authors contributed to the design of the study. F.R. developed the biophysical photoreceptor model; S.I. implemented the biophysical photoreceptor model as a neural network layer and designed and implemented CNNs; J.Z. supervised the implementation of the biophysical photoreceptor model as a neural network layer and the design and implementation of CNNs; G.D.F. and M.B.M supervised retina electrophysiology experiments; J.Z. supervised the overall study. All authors wrote the manuscript.

## Competing interests
The authors declare no competing interests.
