## [Peer Review File · Nature Communications]

Biophysical neural adaptation mechanisms enable artificial neural networks to capture dynamic retinal computationREVIEWER COMMENTS

Reviewer #1 (Remarks to the Author):

This manuscript describes a new model to capture the visual processing within the retina, based on a deep convolutional neural networks (CNN) coupled to a biophysically realistic model of cone processing, for the purposes of capturing visual system adaptation to luminance and contrast conditions. The authors adapt a model of cone processing developed by the Rieke lab as a front end of a CNN model of the retina, use the CNN to fit retinal ganglion cell spike trains recorded in monkey retina (from Rieke lab) and rat retina (from Field lab). They demonstrate that the resulting model can outperform a standard "non-adapting" but otherwise equivalent CNN when tested on lighting conditions that the model was not fit to. Furthermore, they show the adapting model can capture some aspects of the temporal properties of adaptation of ganglion cells.

My opinion is that this is a very important direction for CNNs that are made to explicitly model the [real] visual system. As the authors describe in their intro, CNNs are increasingly used as black-box models to understand visual processing within the brain, but without a realistic front end that captures known effects of cone processing such as adaptation, this approach not only will have poorer performance but also be less biologically interpretable. Thus, this general approach will likely be critical step in using modern machine learning approaches to understand the visual pathway. The cone model itself (with fittable parameters) is an important step already, and here they demonstrate that it can capture neural responses over a wide range of luminance (contrast?) levels in both monkey and rat retina. Furthermore, this manuscript presents high quality, careful work, and also benefits from clear writing and a transparent description of the positive and negative aspects of their results. The potential of this model is clear going forward, and they furthermore use sophisticated and future-looking methods of analysis, including the instantaneous receptive field.

However, while a potentially useful model, this manuscript is lacking in scientific results. The main results validating the model are very modest performance improvements, and a demonstration of the model's ability to capture some known adaptation in retinal ganglion cells: it is not clear what is learned about the retina or visual processing. Along these lines, important aspects of adaptation (arguably the most important), such as change in sensitivity and its dynamics over medium time scales, are would be potentially captured by the model, but are simply left out, as the authors focus on changes in static processing. Finally, while arguably a necessary first step in developing and studying such a model, the application to checkerboard stimulation at constant adaptation states is likely not a useful context to demonstrate the utility or power of the model, and as a result it seems that the exciting aspects of using this model are deferred to future work.

Below I detail these larger concerns/missed opportunities, which are balanced against the overall [great] quality of the current work and its potential for exciting future application.

MAJOR CONCERNS

1. Is this a biophysically correct model of "front-end" adaptation?

One of the two purposes of incorporating a detailed cone model into the front end of a CNN is to properly capture cone adaptation in the front end so that the CNN can model the rest in a more biologically interpretable way. (The other purpose is just so enable the CNN to do a better job at fitting visual system responses regardless of biological interpretation.) But the authors made choices in this manuscript that might undercut this purpose:

A. Changes in sensitivity (the orders of magnitude differences in visual contexts) are arguably the most important role of adaptation. While apparently captured by the cone model (Figure 5), sensitivity in both PR-CNN And vanilla CNN are then adjusted through the normalization layer (and potentially batch norm). I am concerned that taking changes in sensitivity out of the adaptation (cone) model

might continue some of the problems that this approach is meant to solve (correctly attributing adaptation effects). Could the authors clarify how these two aspects of adaptation interact (or do not) in the model and in the retina, and whether this aspect matters for their conclusions?

B. My understanding is that the authors use a cone model, but the lowest luminance condition seems to be largely driven by rods. Likewise, this is also likely the case in the rat experiments for the lower luminance condition. How should we be thinking about this: is a validated regime of the actual cone model? This is a concern since this luminance condition is the one that largely separates the vanilla CNN from the model presented here, and links to concern #2 below.

C. It also raises the question of how important are the details of the cone model (with fittable parameters), versus just having any old adaptation (see below). The particulars of the cone biophysics model are not addressed at all here (in fact its details are relegated to the supplemental), as just a list of equations. What aspects of this model are important to include, or is it best to handle as a black box?

2. Extra spatiotemporal parameters versus valid cone model

To what extent is the success of the cone model here simply a result of adding additional fittable spatiotemporal parameters. While presumably adding parameters anywhere to the model would not be effecting, having parameters that can adjust the temporal kernels in the first layer (as the cone model does) seems to be critical. But how important is it to use the explicit cone model from Angueyra et al (2021) versus any sort of model that can fit adjustable temporal parameters? Can the authors show what is specifically important about the components of the cone model? (At stake is the claim that the model is correctly attributing computation in the CNN by using a biologically realistic cone model: if it is just another set of fittable parameters, this claim is less supported.) One thing that would be reassuring is that the "fittable" parameters of this model matched those determined through the direct cone experiments in Angueyra 2021, or am I thinking about this wrong?

3. Missing dynamics of adaptation

One of the potential main strengths of the cone model is its application to more natural stimuli that have dynamic changes in the adaptation state of the retina, which presumably the cone model can capture. However, this is neither addressed nor validated in the model.

One important question would be whether the PR-CNN would work if the luminance were dynamically changing as it would in more natural visual contexts? In this sense, the results seem implicitly limited by the stimulation condition tested: checkerboard stimuli at constant luminance/contrast.

4. Role in coding?

One potential missed opportunity is to describe whether the PR-CNN confers (or captures) any coding advantages rather than simply reproducing ganglion cell responses better. Adaptation has a clear functional role in adjustment of sensitivity, but this seems like it can be handled by the normalization layer without any PR-CNN. If we were designing an ANN to process natural vision (rather than simply explaining neural responses), would we want to include something like a cone adaptation model? Does it have a functional purpose, or is it viewed by the authors as simply a biological constraint?

5. Number of cells used in primate

The small number of cells used in this study also might be a limitation — although I could be convinced otherwise if more was said about this. I worry both about the dependence on unequal sampling of cell types, and also the amount of data necessary to train the CNN, and whether it overfits to the particular details of the 37 cells.

MINOR

— Abstract (line 2): conventional ANNs are generally not predicting neural data, so the scope implied

is a bit confusing

- lines 54-55: Seems one more paper should be added to the retina-DNN list (that I know of): Batty, E. et al. Multilayer Recurrent Network Models of Primate Retinal Ganglion Cell Responses. International Conference on Learning Representations (2016)
- lines 106-7: This is written in a way that makes it sound like rods dominate at all three light levels. (easy fix). Later, it would be useful to say whether this is the case in the rat retina.
- How were 37 cells selected? (simply based on the quality of the spike sorting?)
- Should we be worried that the PR-CNN actually performs worse at high luminance? From the figure (Fig 2C) it seems it cannot capture the largest responses. Can more be said about this?
- Is model performance calculated as a mean over all neurons? (check methods)
- Could more be said about the motivation for including batch normalization (lines 133-134), which I do not think will be familiar to a general audience. In particular, my understanding it simply determined a scaling (like normalization layer) and offset within layers of the CNN, but is frozen at test time (and thus amounts to a simple shift in weights and biases). Is it "frozen" for each luminance condition differently?
- I would appreciate more description of the Angueyra model in the supplemental: at least what the variables represent.

Reviewer #2 (Remarks to the Author):

Summary

Idrees et al introduce a convolutional neural network with a photoreceptor model front-end that captures retinal adaptation across light levels. They fit this network to multielectrode-array data from isolated macaque and rat retinas under white noise stimulation at different light levels. The paper focuses on comparing model performance of networks with or without photoreceptor biophysics to conclude that the inclusion of an adaptive front-end generalizes well across light levels. One key highlight (introduced rather late in the paper) is that models of rat retinas trained at photopic light levels can predict well at scotopic conditions. Although this work intersects the domains of retinal neuroscience and computer vision, it offers only incremental advancements in both. While the premise of the study is interesting, the authors never go beyond a model implementation to show how photoreceptor adaptation can contribute to either retinal or computer vision.

Recent observations from the Rieke lab (Anguera et al 2022 and Yu et al 2022) showcase that cone adaptation is relevant for natural vision at timescales of single fixations and how this adaptation can lead to downstream (nonlinear) effects in the retinal output. The choice of white noise stimulation by the authors washes out all the nuance previously shown with natural visual inputs. White-noise stimuli could potentially be analyzed to reach similar conclusions with methods simpler than fitting CNNs. The authors' choice to focus on white noise is particularly unfortunate because it makes the fitted models hard to interrogate further. For example, it has been shown that models trained on natural scenes but not noise can reproduce a wide variety of known phenomena observed with artificial stimuli (Maheswaranathan et al 2019).

The authors' adaptive photoreceptor network is compared with simple CNNs (with BatchNorm) that interpolate photoreceptor dynamics when trained on two extreme light levels (Figure 3). This finding overshadows the impact of this study in the field of computer vision because it reduces the prediction problem in providing the right training data. As the authors also note (lines 300-312), the photoreceptor layer could be replaced by an alternative dynamic layer and modern vision architectures (e.g., with attention mechanisms) offer such highly adaptive solutions. Focusing more on what the biophysical photoreceptor layer offers relative to a generic adaptive layer is a key question to answer. For example, having to fit only four photoreceptor parameters instead of hundreds of generic ones can lead to better generalization with less data.

Overall, the message/conclusion of the study is plausible, but all claims made are only loosely supported by the data. Key methodological details are missing from the text, and the (limited) data are only poorly explored. This data limitation could be overcome with stronger conceptual developments or applications that the authors suggest (lines 92-96).

Major comments:

- Experimental considerations

The major part of this study's results comes from a single macaque retina recording. Data quality comes to question given that only 37 good cells came out of a dense 512-electrode array recording, which previously other labs showcased that can produce at least hundreds of simultaneously recorded cells (e.g., Rhoades et al 2019). Light responses in the macaque retina are extremely sensitive to preparation conditions; there are empirical standards to assess light sensitivity of macaque retinal tissue (e.g., Rieke, Manookin lab papers). Can the authors provide evidence that the retina was in good condition? How well did cells respond across trials of the test (fixed) white-noise stimulus? Finally, I assume that the retinal piece the authors are analyzing comes from the peripheral macaque retina, but this detail is missing.

Since the generalization from photopic to scotopic levels is impressive (Figure 7), the study could strengthen its most significant result by analyzing previously recorded data. For example, the Field et al 2009 study, whose first author is also part of this manuscript, includes photopic and scotopic white noise measurements in populations of macaque RGCs. Would the rat results generalize in the macaque condition?

Given the limited data, a thorough exploration of the macaque dataset could strengthen the impact of the paper. It is clear from multiple plots that the reported performance is highly variable across cells. Is this variability somehow related to the recorded cell types (midget/parasol)?

- Model architecture and baseline comparisons

While the study is focused on biophysical properties of the photoreceptor layer, the remaining architecture is quite far from the actual retinal biophysics. How was the architecture chosen? The authors state a grid search, but what was the objective and which data were used? Starting from white-noise stimulation, the stixel size used was rather large (140 μm for macaque and 252 μm for rat). At this size length, one pixel is at least as large as a midget ganglion cell receptive field center, and probably consists of multiple bipolar cells. Thus, a single convolutional layer may potentially capture all the operations performed by the three-layered architecture the authors used. The stride used for the first CNN layer (9 pixels) requires integration of information >1 mm which is rather large if we were to interpret the first layer as the photoreceptor or bipolar cell mosaic. Such long-range information integration only happens for wide-field amacrine cells.

Although the paper frames CNNs in the center, a multi-layered CNN architecture is not a prerequisite to train models for RGCs in parallel (e.g., Pillow et al 2008). An alternative that could still reveal photoreceptor effects is to train multiple GLMs (one per cell), all receiving shared input from the adaptive photoreceptor layer. It is possible that the authors can fit such individual models with even less parameters. For example, using (per cell) a 5×5 spatial filter, 10 temporal parameters (using basis functions) plus 2-3 parameters for the output nonlinearity results in 1400-1500 parameters for all 37 cells. It is clear that the CNN approach may be more data-efficient for larger cell counts, but does weight-sharing in the authors' multi-layered CNN architecture actually reduce this individual model parameter estimate by a lot?

- Model comparison and statistical methods

The main model evaluation method, fraction of explainable variance (FEV), is actually left unexplained throughout the whole manuscript (despite a reference in line 127). Is this an R-squared-based measure compared to trial-to-trial variability? In any case, the calculation should be reported in the methods. Are cells with negative FEV cells with extremely bad predictions or cells for which the firing was unreliable throughout the recording? Were there any cell inclusion criteria for fitting the model as there were for follow-up analyses (Figure 5)? Did cells show consistent responses over the course of the experiment?

How well do ganglion cell responses themselves predict responses at different light levels? If one computes FEV for 3R* using the 30R* responses, how much does one expect to predict? If this number is quite large, I would argue that it becomes a challenge to evaluate whether the photoreceptor layer adds much using these data. This is because there may be little differences in the test set to begin with. This baseline analysis is of particular interest in the case of Figure 7, where the two models compared have drastically different predictions.

The most concerning observation is that the authors used a sample t-test to obtain their main result (Figure 3), in contrast to the rest of the paper where they use appropriately use non-parametric statistics. Do signed-rank Wilcoxon tests reveal statistical significance between model performance?

Minor comments:

- The Anguerya et al. study (from which the photoreceptor model comes from) is now published in the Journal of Neuroscience (2022). The authors should update the citation details.
- It is unclear whether the length of movie segments used for training is 120 (line 120) or 180 (line 442)? Such a length is a bit unusual for describing temporal filters of ganglion cells. 120 frames at 8 ms each is 960 ms, but primate RGC filters require less than half of that length to capture light responses (this is also seen in the authors' data in Figure 4). Is this long duration to allow for the photoreceptor model to integrate over the recent intensities? A long filter requires more data to fit and could potentially lead to overfitting of temporal components for the baseline CNN model (as seen in Figure 4b).
- Figures 2,6 and 8. Are the RGC responses shown averages of repeated noise segments or smoothed single-trial responses? Because they mention repeated segments of five to ten seconds long, it would be informative to compare model responses across the whole segment and not just two seconds.
- How were rod isomerization values calculated?
- How do the estimated photocurrents differ for scotopic and photopic levels in the rat retina? The differences should be more dramatic compared to the analysis of Figure 6f.
- Are the rat retina results symmetric over the two recordings? If trained on Retina B, the photoreceptor model should also generalize to Retina A.
- Units of firing rates in single-cell examples. In Figure 2, 6 and 7, it is unclear whether the firing rate is normalized or not. A normalized measure typically has no unit (in the Figures, instead, spikes/s is given). For these plots, it would be helpful to examine the actual response magnitude (in spikes/s) and also where the zero baseline lies.
- The authors should explain how they decompose receptive fields into spatial and temporal components. Do they just use the stixel frame/timecourse where the highest absolute value occurs?
- The rat recordings contained only OFF cells. Is there a reason that ON cells were not included?

Reviewer #3 (Remarks to the Author):

The paper discusses a novel convolutional neural network (CNN) for vision that integrates a biophysical model of the phototransduction cascade within the photoreceptors. This novel model, referred to as the photoreceptor-CNN (PR-CNN), significantly enhances the prediction of retinal ganglion cell (RGC) responses across varying light levels, outperforming conventional CNN models. I believe this paper represents an important advancement in artificial neural network (ANN) models of the visual system. While the primary focus of the paper lies in demonstrating the benefits of incorporating a realistic photoreceptor layer for predicting RGC responses, its implications extend far beyond, potentially impacting our models of the entire visual system and downstream higher-order visual areas. This paper is a great contribution to both neuroscience and AI, and I support its publication.

However, I have a few comments that need to be addressed to enhance the clarity of the paper:

Major comments:

1- Although the advantages of the bio-realistic photoreceptor layer in predicting RGC responses are evident, as a reader, I found it challenging to grasp the mechanistic model of photoreceptors. While the authors do provide the equations of the photoreceptor in the appendix, an introduction to the model and some intuitive explanation within the main text would greatly help.

2- Considering that the photoreceptor model involves dynamic nonlinearities, it would be beneficial to explore whether increasing the depth of the regular CNN model leads to better predictions of RGC responses. Is there a specific dynamical motif that multiple (recurrent) layers might fail to learn and represent? Is the improvement in performance attributable to the larger capacity of the model with the photoreceptor layer or the specific inductive biases of the photoreceptor layer itself? Conducting more detailed experimentation, such as ablations, involving different trainable components of the photoreceptor layer could provide valuable insights, particularly in understanding the computations required for context-dependent visual processing. Addressing this concern is closely related to the need for an intuitive understanding of the photoreceptor layer to comprehend the underlying reasons behind the obtained results.

3- The performance of PR-CNN in the $30 \text{ R}^* \text{receptor}^{-1} \text{s}^{-1}$ condition appears to be lower than that of the CNN model. It would be beneficial if the authors could provide an explanation for this discrepancy and determine whether it indicates a systematic change.

Minor comments:

1- Figure 5 requires color bars to distinguish different bins clearly.

2- In Figure 4, specifically in panels b and c of the bottom row, the use of different colors needs clarification.

3- Line 236 mentions, "This change in kinetics could already be observed at the output of the photoreceptor layer." It is not entirely clear what changes in kinetics in Figure 6f should be observed. The authors should clarify this point for better understanding.

Reviewer #4 (Remarks to the Author):

Summary of the manuscript:

The authors present a novel convolutional layer to add as a first entry layer before traditional convolutional neural networks models (CNNs) of the retina.

Their main claims are that this layer, called Photoreceptor Convolutional Layer, which is based on a biophysical model previously published, are:

- 1- that the new model can correctly capture changes in sensitivity, which they call "gain".
- 2- that the new model captures temporal dynamics that change according to the intensities of the input images
- 3- that these properties allow the model to generalize to different ranges of intensity, outperforming traditional CNNs

They perform modeling on retinal ganglion cells (RGC) on a new dataset from one monkey retina and in two datasets already published from two rats retina.

The research is novel and pertinent. If proven generally successful, it may be included as a first layer in artificial intelligence (AI) vision processing models. This will be a good advancement in the field of research interested in obtaining biologically plausible and realistic models, in order to gain insight from the parameters and inner layers of models, in contrast to the more general "black box" approach that deep neural networks in AI modeling usually have.

However, I think the authors could have done a much better job to prove the advantages, and specially to characterize the properties of their new neural network with a biophysically inspired input layer.

Major comments:

1) INTERPOLATION VS EXTRAPOLATION: My first immediate question when reading the manuscript was, why do the authors train in the upper and middle range of intensities and test in the lower, instead of training upper and lower ones and testing in the middle? I was expecting, as is usual for AI models, that the standard CNN they were trying would not extrapolate to "out of training range" intensity values, as the authors show. However, they later show that they trained the model as I expected, and that it performs as good as the new model they propose. Then, all the rest of the article is focused on the capability of the model to generalize when extrapolating, and comparing its performance with the classical CNN. This is somehow killing their argument that their model is better "per se". In my view, is that they just trained the classical CNN with the wrong dataset. However, if their claim was focused on that the Photoreceptor layer can extrapolate to different intensity ranges, this is a different story. The authors should make this clear and more transparent.

2) IS CNN INTRINSICALLY WORSE?: The performance of the models when interpolating is similar (Figure 3d). Then, it would be interesting to compare the parameter convergence of the models to understand what is failing for the CNN when extrapolating, or what characteristics of the photoreceptor layer is modeled into the CNN that works better (since the best PR-CNN is comparable to the CNN). Is the CNN of figure 3d able to capture the claims for the PR-CNN shown in Figures 5 and 6? If the answer is yes, then having a PR-CNN does not imply that this type of model can "only" capture points 1- and 2- in the summary above. The authors should explicitly compare the number of parameters of each model. Is it that the PR-CNN is better because it has more parameters? I doubt this, but a clarification can help. Related, how would Figure 6d look for the interpolated CNN. Can this CNN capture time kinetics even without a PR-CNN?

3) WHAT DID WE LEARN?: The performance in general is poor at this stage of the model when extrapolating (around 50%), compared to the more than 80% achieved when training and test are the same. So, apart from the fact that a photoreceptor layer can extrapolate to other intensity ranges, is there a difference between the three models learned when testing at different values? As I

understand, they retrained each time. So, can we compare the parameters obtained for the PR layer (they are only 4, and they could maybe easily studied/interpreted). Related, in line 324 authors say: "the biophysical photoreceptor model we use in the photoreceptor layer while complex, has parameters that map directly onto the biology, providing an opportunity for investigating the role of photoreceptor adaptation in the retina." (plus following sentences). Can the authors show/interpret this with their fitted parameter models?

4) RODS vs CONES. The rationale to go from rods to cones is not smooth. First, when they introduce the light levels, they do not motivate it. Why would they want to study rod dynamics and not cone dynamics? Is it because it will be easier to treat only one active photoreceptor and capture its dynamics? Then, the reference the authors mention, where the PR model is built (Angueyra et al., 2021), presents data ONLY for cones (for monkeys in another intensity regime). Then, wouldn't it be necessary to make at least a comment on this transition? Is it reasonable to use a model fit for cones directly for rods? Should this be later checked experimentally? Regarding the methods, what is the rationale for fixing 8 parameters?, what is the bibliographic reference to find these fixed values? I understand the code will be available, but a table with the values chosen may help. More so if there are two different types of photoreceptors fitted in the manuscript. Is there an intrinsic difference between them? If the model of Angueyra et al., 2021 (still in preprint form) is the first to describe this, then in this manuscript, more effort could be put to show the difference between cones and rods in the model output and/or parameters. I understand that the authors want to put the equations in supplementary material, but some diagram like Figure in Fig6A of Angueyra et al., 2021 will be of much help.

Finally, do Figure 5 and 6 results hold for photopic levels (or maybe show more drastic differences?). Or cones and rods present different dynamics for this data? (also monkey and rat data may differ). I see in Fig7a that the fit is bad. (maybe too bad), so maybe this quantification cannot be done? Could the authors confirm that this is the best possible fit for CNN without a photoreceptor layer? Or is it that I am misinterpreting Fig 7a (it may be because this section is a bit disorganized and difficult to follow). I think Fig7 result will be very much more impactful, if all my comments above are addressed. I had to make a lot of effort to understand what was presented.

5) SINGLE CELL DATA: Although the main claim of the model performing better is in Figure 3 showed as percentages in the population with box plots and outliers, a figure with the individual performances comparison for individual cells is missing and is a common practice to assess the goodness of a model (see outliers, general trends, etc.). Other measures may be used also, to see if the PR-CNN model is doing better for just a subset of cells, or if it makes it better for all of them.

Minor comments:

1) The authors mention "changing input conditions", "sensitivity", "gain", but are vague in general, until getting to the results. It may help to be more specific onto which type of changes they refer to, and use a single nomenclature across the manuscript.

2) The way data is first presented is confusing. The percentages in the main text do not match the percentages one first encounters in Figure 2. It does not help that one of the population % matches the example given (78%). This can be easily changed by reordering figures or text.

3) The " R^* receptor-1s-1" is cumbersome to read each time the authors mention the light levels, and could be easily replaced by $R^*/ph.s$ or directly R^* if defined the first time it appears. It is cumbersome to read it. It could also be avoided by naming H M and L the high, medium and low levels of stimulation. Same for the Layer Normalization Layer: could be said Normalization Layer.

4) Maybe I am wrong, but it seems that here there is an extra layer compared to Deep Retina (the 18

channel one). Is this correct? Was this necessary here for the Photoreceptor layer to work better? If this layer is not present, PR-CNN does not work, or is it just worse?

5) "Normalization by mean and variance" sounds strange, you mean z-scoring? In line 120 says it has a similar effect to z-scoring, then I do not understand the difference.

6) A formula for the FEV will be helpful in the methods. Especially for interpreting the negative values.

7) Line 138 "it predicted the same output for a given movie segment". I do not see this explicitly proven later in the manuscript, although somehow Fig 6c points on that direction.

8) Line 141, Input conversion, the authors mean just "different input ranges"?

9) There is an inconsistency between the frequency reported for the movie (60/30Hz), and the frame length used for the modeling (8ms).

Does this mean that the movie frames were stepped in subframes? How did the authors treat the data?

10) When taking out some data, a plot of the performance of the cells could be helpful, to understand the rationale of the threshold selected (Line 210), since values at around 50% are quite low to be "more reliable".

11) I would expect Figure 4a to be a step function, but I see diagonal lines. In the case of the CNN in 2b, an explanation to understand why is much less smooth than the PR-CNN could be helpful. Finally, the method to decompose space and time is not explicitly specified in the methods.

12) In figure 6a, if a curve is normalized, then shouldn't it be in arbitrary units, and below 1?

13) the differences in Fig6f do not stand out, I would zoom in or include an inset.

14) The fact that the gradients are so smooth in PR-CNN does not come straightforward. This seems an advantage of the model.

15) "We leave that analysis for future work". This type of phrase could be ruled out. It makes it feel that this should be done now in this manuscript. Same for "future studies will", "future work will" change it for "may".

16) 10% refractory period contamination seems a cortical like extracellular recording. I think that the standards for MEA in vitro are much better. Some papers reporting less than 1% at 2ms windows. Authors should revise this to explain why such high values are permitted.

17) Labels in Fig6a is superimposed with numbers.

18) Something like Supplementary Figure 1 (with a better description), may help very well to introduce/explain the photopic vs scotopic experiment if included in Figure 7.

**Responses to reviewer comments on:
“Biophysical neural adaptation mechanisms enable deep learning models to capture
dynamic retinal computation”**

Idrees, Rieke, Field and Zylberberg

We thank the reviewers for their constructive comments that identified several areas for improvement in our initial manuscript. Their most substantial critiques were that: 1) the initial submission lacked results showing performance of the proposed model for more complex stimuli such as natural scenes; 2) the initial submission lacked ablation experiments to understand mechanisms that gave rise to observed performance gains with the photoreceptor-CNN model.

In response to these critiques, we collected new experimental data from macaque retina stimulated with natural scenes, at higher spatial resolution than in our initial submission. With these new data, we observe that the photoreceptor-CNN model substantially outperforms regular CNN models at predicting responses to natural scene stimuli, even at the training light level. For contrast, we do not see such performance gains at the training light level for white noise stimuli (as in our original submission). We attribute this difference to the temporal correlation structure and diversity of contrast values present in natural scene movies which activate dynamic photoreceptor adaptation even at steady-state mean global light levels. Moreover we have undertaken ablation experiments demonstrating that the nonlinear adaptation and explicit feedback mechanisms of the biophysical model are essential for the observed performance gains. With these new data, we have added two new result sections and have substantially reorganized the manuscript. We believe that it is much improved from our initial submission. The reviewers' specific critiques (black text) and our responses are outlined below (green text).

REVIEWER COMMENTS

Reviewer #1

This manuscript describes a new model to capture the visual processing within the retina, based on a deep convolutional neural networks (CNN) coupled to a biophysically realistic model of cone processing, for the purposes of capturing visual system adaptation to luminance and contrast conditions. The authors adapt a model of cone processing developed by the Rieke lab as a front end of a CNN model of the retina, use the CNN to fit retinal ganglion cell spike trains recorded in monkey retina (from Rieke lab) and rat retina (from Field lab). They demonstrate that the resulting model can outperform a standard "non-adapting" but otherwise equivalent CNN when tested on lighting conditions that the model was not fit to. Furthermore, they show the adapting model can capture some aspects of the temporal properties of adaptation of ganglion cells.

My opinion is that this is a very important direction for CNNs that are made to explicitly model the [real] visual system. As the authors describe in their intro, CNNs are increasingly used as black-box models to understand visual processing within the brain, but without a realistic front end that captures known effects of cone processing such as adaptation, this approach not only

will have poorer performance but also be less biologically interpretable. Thus, this general approach will likely be critical step in using modern machine learning approaches to understand the visual pathway. The cone model itself (with fittable parameters) is an important step already, and here they demonstrate that it can capture neural responses over a wide range of luminance (contrast?) levels in both monkey and rat retina. Furthermore, this manuscript presents high quality, careful work, and also benefits from clear writing and a transparent description of the positive and negative aspects of their results. The potential of this model is clear going forward, and they furthermore use sophisticated and future-looking methods of analysis, including the instantaneous receptive field.

However, while a potentially useful model, this manuscript is lacking in scientific results. The main results validating the model are very modest performance improvements, and a demonstration of the model's ability to capture some known adaptation in retinal ganglion cells: it is not clear what is learned about the retina or visual processing. Along these lines, important aspects of adaptation (arguably the most important), such as change in sensitivity and its dynamics over medium time scales, are would be potentially captured by the model, but are simply left out, as the authors focus on changes in static processing. Finally, while arguably a necessary first step in developing and studying such a model, the application to checkerboard stimulation at constant adaptation states is likely not a useful context to demonstrate the utility or power of the model, and as a result it seems that the exciting aspects of using this model are deferred to future work.

Below I detail these larger concerns/missed opportunities, which are balanced against the overall [great] quality of the current work and its potential for exciting future application.

We appreciate the valuable suggestions from the reviewer. In response, we conducted additional electrophysiology recordings and associated modeling experiments using primate retina data, encompassing RGC responses to naturalistic movies and checkerboard stimuli at a finer spatial scale. Our findings indicate that even at a single light level (at which the models are trained and tested), the photoreceptor-CNN model consistently outperforms the conventional CNN in predicting RGC responses to naturalistic movie stimuli (new Figure 2, 3a, Supplementary Fig. 1). This was not apparent in our previous experiments (now Fig. 5) and we attribute this difference to the temporal correlations and diversity of contrast levels present in natural scene stimuli that may activate photoreceptor dynamic adaptation at a steady-state mean light level.

Additionally, to understand whether nonlinear adaptation in the photoreceptor layer underscores this performance gain, we performed an ablation experiment where the nonlinear biophysical model was replaced by a linear photoreceptor model. The linear photoreceptor model performance was similar to the conventional CNN (new Fig. 3b). This suggests that nonlinearities and feedback mechanisms in photoreceptor processing are a key component impacting retinal responses to stimulation – even at a steady state mean light level.

While our model performance is not yet perfect (which would correspond to 100% FEV), the photoreceptor input layer does substantially improve it: in experiments where we test performance at held-out light levels, it can increase performance by up to a factor of 2 as compared to vanilla CNNs. In this most extreme case, the vanilla CNN achieves 24% FEV, and

that increases to 54% with the inclusion of the photoreceptor layer (Fig. 5c, formerly Fig. 3d). Moreover, in our new experiments with naturalistic movie stimuli, the PR layer can improve the CNN performance by ~29% as compared to vanilla CNNs without the PR layer (new Fig. 3a: performance improves from 38% FEV to 49% FEV). This performance gain is achieved even when the model is trained and tested at the same mean light level.

Major comments

1. Is this a biophysically correct model of "front-end" adaptation?

One of the two purposes of incorporating a detailed cone model into the front end of a CNN is to properly capture cone adaptation in the front end so that the CNN can model the rest in a more biologically interpretable way. (The other purpose is just to enable the CNN to do a better job at fitting visual system responses regardless of biological interpretation.) But the authors made choices in this manuscript that might undercut this purpose:

A. Changes in sensitivity (the orders of magnitude differences in visual contexts) are arguably the most important role of adaptation. While apparently captured by the cone model (Figure 5), sensitivity in both PR-CNN And vanilla CNN are then adjusted through the normalization layer (and potentially batch norm). I am concerned that taking changes in sensitivity out of the adaptation (cone) model might continue some of the problems that this approach is meant to solve (correctly attributing adaptation effects). Could the authors clarify how these two aspects of adaptation interact (or do not) in the model and in the retina, and whether this aspect matters for their conclusions?

Layer normalization and batch normalization are operations that re-scale the magnitudes of the inputs to the CNN, or its hidden unit activations. As a result, they can help to compensate for changes in input magnitude – e.g., due to changing luminance levels. Notably, in the vanilla CNN, if Layer Norm is removed, the amplitude of predicted responses at the held-out test light level is substantially incorrect, and the CNN performance is very poor. While Layer Norm equips vanilla CNNs to re-scale their inputs (and outputs) as the luminance changes, it does not enable the CNNs to capture more complicated changes in the way the retina adapts to changing light levels: e.g., dynamic changes in sensitivity resulting from rapid luminance changes as in natural scenes (Fig. 2a) and changes in response kinetics (Fig. 4c) . For contrast, the photoreceptor model does change its sensitivity dynamically (Fig. 2b) and its kinetics (Figs. 4d, 6), as the luminance changes, enabling it to better emulate retinal adaptation.

A technical reason for including Layer Norm and Batch Norm in our models is that they stabilize model training and facilitate quicker convergence in fewer training epochs. During training, the distribution of inputs to a layer undergoes changes as the network's parameters are updated, leading to what is known as the internal covariate shift — a phenomenon that hampers model convergence and introduces instabilities. Layer Norm and Batch Norm both help to address this issue by normalizing the input distribution to each layer. In the photoreceptor-CNN model, this normalization specifically at the PR output is crucial due to substantial differences in scale between the biophysical model's

parameters and the downstream CNN weights. Removing these normalization layers hinders the photoreceptor-CNN model's convergence.

As helpful as it may be in training the models, the reviewer pointed out that adding Layer Norm may also introduce a challenge in isolating adapting processes within the PR model. This is correct for steady-state sensitivity changes, such as those that would occur across steady-state changes in light levels. This would make investigating the role of photoreceptor adaptation across steady-state light changes more challenging as one will have to also account for the sensitivity changes mitigated by Layer Norm. However, it is important to note that dynamic sensitivity changes, such as those that occur when local intensity changes rapidly, will still remain attributable to the photoreceptor layer since Layer Norm cannot capture such local changes in sensitivity that occur within a single movie segment. The new experiments we conducted in these revisions somewhat address this concern by modeling RGC responses at a single light level (new Figs. 2, 3, Supplementary Fig. 1). In these models, Layer Norm would have minimal impact in mitigating dynamic sensitivity changes triggered by fluctuations in pixel intensities within a training sample (80-frame movie segment).

We have now included this information in Discussion (lines 342-349).

B. My understanding is that the authors use a cone model, but the lowest luminance condition seems to be largely driven by rods. Likewise, this is also likely the case in the rat experiments for the lower luminance condition. How should we be thinking about this: is a validated regime of the actual cone model? This is a concern since this luminance condition is the one that largely separates the vanilla CNN from the model presented here, and links to concern #2 below.

The underlying model is a published biophysical model of the phototransduction cascade. Depending on the values of the model parameters, it can be configured either as a cone photoreceptor or rod photoreceptor. Except for the rat experiment at photopic light level, the initial values of the photoreceptor model parameters were set to experimental fits to primate rod photoreceptors determined as part of other studies in the lab of Fred Rieke (Chen et al., 2024). We used mouse cone and rod photoreceptor model values (Chen et al., 2024) for modeling photopic and scotopic light level respectively in rat experiments. We have also included these values in Supplementary Table 1.

C. It also raises the question of how important is the details of the cone model (with fittable parameters), versus just having any old adaptation (see below). The particulars of the cone biophysics model are not addressed at all here (in fact its details are relegated to the supplemental), as just a list of equations. What aspects of this model are important to include, or is it best to handle as a black box?

To address this question, we performed an additional experiment (Fig. 3b) where we replaced the biophysical model with a simpler alternative model, capable of accommodating linear adaptive effects. The performance of the resultant model was

lower than the full biophysical model, which incorporates nonlinear adaptive mechanisms and explicit feedback (which depends on calcium accumulation in the photoreceptor) which are absent in the simpler model. This outcome emphasizes that the nonlinear adaptation captured by the biophysical model, and its ability to appropriately modulate that adaptation, are key contributors to the observed performance gains. These results are presented in a new section (lines 150-182).

2. Extra spatiotemporal parameters versus valid cone model

To what extent is the success of the cone model here simply a result of adding additional fittable spatiotemporal parameters. While presumably adding parameters anywhere to the model would not be effecting, having parameters that can adjust the temporal kernels in the first layer (as the cone model does) seems to be critical. But how important is it to use the explicit cone model from Angueyra et al (2021) versus any sort of model that can fit adjustable temporal parameters? Can the authors show what is specifically important about the components on the cone model? (At stake is the claim that the model is correctly attributing computation in the CNN by using a biologically realistic cone model: if it is just another set of fittable parameters, this claim is less supported.) One thing that would be reassuring is that the "fittable" parameters of this model matched those determined through the direct cone experiments in Angueyra 2021, or am I thinking about this wrong?

We don't intend to claim that all of the biophysical details of the photoreceptor model are critical to its performance. Rather, our modeling work highlights that the adaptive mechanisms captured by that photoreceptor model enhance CNN performance. We acknowledge that if other empirical models can capture similar input-output mappings, they may perform similarly in predicting RGC activation even if they do not match the mechanistic details of the photoreceptor: at the same time, models that do not match the biophysical details may be of more limited use in investigating the mechanistic foundations of vision (lines 371-381).

The PR-CNN model explicitly lacks other forms of dynamic adaptation that are present downstream of the phototransduction cascade such as synaptic depletion/facilitation. When training the PR-CNN model end-to-end, including the PR layer parameters, it is likely that some of these downstream adaptive effects are captured by the trained PR model due to its ability to represent a range of different dynamics. While this approach yields an optimal predictor, the interpretability of the learned PR models' values may be compromised. To address this concern, we conducted additional experiments where the PR layer's parameters were fixed to their experimentally-constrained values (from separate experiments on primate rod cells, performed by Fred Rieke). The model with fixed PR parameters had similar performance as the trainable PR-CNN (refer to new Fig. 3c; fixed PR-CNN = 49%; trainable PR-CNN = 49%). This also shows that we may not need to train the photoreceptor layer at all: we may in many cases be able to leave it with the experimentally measured parameters.

Additionally, it is worth noting that the rod parameter values used here were obtained from experimental fits to rods from a different macaque retina than the one from which our RGC recordings were obtained: this emphasizes the potential for biophysically-constrained models to help generalize between different retinæ.

3. Missing dynamics of adaptation

One of the potential main strengths of the cone model is its application to more natural stimuli that have dynamic changes in the adaptation state of the retina, which presumably the cone model can capture. However, this neither addressed nor validated in the model.

One important question would be whether the PR-CNN would work if the luminance were dynamically changing as it would in more natural visual contexts? In this sense, the results seem implicitly limited by the stimulation condition tested: checkerboard stimuli at constant luminance/contrast.

This is a very important point and is a study that we had originally left for future work. To address the reviewer's concern, we have performed new experiments that include naturalistic stimuli, and included them in the revised manuscript (new Figs. 2,3, Supplementary Fig. 1). In these experiments, we first recorded macaque RGC activity in response to naturalistic movies generated by moving natural images from Van Hateren's dataset across the retina so as to follow simulated eye movement trajectories. We modeled these RGC responses at a single light level. The PR-CNN model shows substantial performance gains as compared to the vanilla CNN model (PR-CNN = 49% FEV; vanilla CNN = 38% FEV), indicating the photoreceptor model's ability to capture adaptation during dynamic luminance changes such as those induced during eye movements. These adaptive effects were not captured by the model where the biophysical PR layer was substituted with a linear PR layer (Fig. 3b; linear PR-CNN = 37% FEV).

These results are now included as the first two sections of the revised manuscript.

4. Role in coding?

One potential missed opportunity is to describe whether the PR-CNN confers (or captures) any coding advantages rather than simply reproducing ganglion cell responses better. Adaptation has a clear functional role in adjustment of sensitivity, but this seems like it can be handled by the normalization layer without any PR-CNN. If we were designing an ANN to process natural vision (rather than simply explaining neural responses), would we want to include something like a cone adaptation model? Does it have a functional purpose, or viewed by the authors as simply a biological constraint?

While we acknowledge the missed opportunity in the current manuscript, we are actively pursuing this avenue as a separate project. That project has a fairly broad scope, involving benchmarking the bio-inspired CNN against multiple computer vision datasets. However, it deviates from the primary focus of this paper, which is demonstrating the enhancement of neural predictors for the retina through photoreceptor models.

5. Number of cells used in primate

The small number of cells used in this study also might be a limitation — although I could be convinced otherwise if more was said about this. I worry both about the dependence on unequal sampling of cell types, and also the amount of data necessary to train the CNN, and whether it overfits to the particular details of the 37 cells.

In our new experiments (new Figs. 2,3, Supplementary Fig. 1), we modeled 57 RGCs from a new primate retina. These RGCs include 27 ON parasol cells and 30 OFF parasol cells. While the number of spike sorted cells were much higher, we only selected the cells (1) that passed the spike sorting quality thresholds and (2) that were reliably tracked across the different experimental conditions. No further selection criteria were applied. In the previous experiments (now Figs. 4, 5), we again used only a subset of recorded RGCs that passed the above 2 criteria. These contained 30 ON RGCs (2 parasol + 28 midget) and 7 OFF RGCs (5 parasol + 2 midget).

While incorporating the photoreceptor model is an initial step, our overarching objective is to leverage CNNs as a scaffolding into which we can incorporate other models of biologically known mechanisms, so as to build a trainable yet bio-realistic model of the visual system.

Minor comments

— Abstract (line 2): conventional ANNs are generally not predicting neural data, so the scope implied is a bit confusing

We are referring to ANNs such as CNNs that are now increasingly being used to predict neural data.

— lines 54-55: Seems one more paper should be added to the retina-DNN list (that I know of): Batty, E. et al. Multilayer Recurrent Network Models of Primate Retinal Ganglion Cell Responses. International Conference on Learning Representations (2016)

Thank you for pointing it out. We have now included this paper.

— lines 106-7: This is written in a way that makes it sound like rods dominate at all three light levels. (easy fix). Later, it would be useful to say whether this is the case in the rat retina.

We don't compare primate and rat recordings at these light levels. We have now added more information. Please see lines 195-199

— How were 37 cells selected? (simply based on the quality of the spike sorting?)

They were selected based on spike sorting quality and the ability to track them across the light levels. We have clarified this. Please see lines 426, 463, 497.

— Should we be worried that the PR-CNN actually performs worse at high luminance? From the figure (Fig 2C) it seems it cannot capture the largest responses. Can more be said about this?

This is not true for all RGCs. Individual RGCs show variations. We have now included a figure in the supplementary that shows pairwise performance comparisons for each RGC across light levels (see Supplementary Figure 2).

— Is model performance calculated as a mean over all neurons? (check methods)

It is calculated as median over all the neurons. We have now made this more explicit in the main text (line 100) and in the Methods (line 641).

— Could more be said about the motivation for including batch normalization (lines 133-134), which I do not think will be familiar to a general audience. In particular, my understanding it simply determined a scaling (like normalization layer) and offset within layers of the CNN, but is frozen at test time (and thus amounts to a simple shift in weights and biases). Is it "frozen" for each luminance condition differently?

During the training phase, the batch normalization layer first z-scores its inputs, which are the outputs of the convolution operation. The normalization statistics are computed based on the batch statistics, enforcing a 0 mean and unit variance for the data. Subsequently, batch normalization introduces trainable shift and scale parameters, systematically adjusting all weights and biases in the CNN layer. In addition, Batch Norm saves two non-trainable parameters: the running mean and variance of the data that are used for normalization during the test phase. This normalization process mitigates extreme parameter values that may lead to exploding or vanishing gradients. The scale and shift parameters allow the model to accommodate variations in feature magnitudes across different layers and activations. Learning the optimal scale and shift parameters enables the model to adapt to inherent scale differences, resulting in improved and faster convergence.

During the test phase, Batch Norm uses the non-trainable moving mean and variance that it saved during the training phase, to normalize its inputs i.e. the outputs of the convolutional layer. It then scales and shifts the normalized input using the two parameters that were learnt during the training phase.

In the training phase, each batch contains 100+ training examples from the light level(s) the model is being trained at. However, at the input, we have Layer Norm which z-scores each training example based on statistics of that single training example (the layer norm in this case operates the same at train and test phase). This approach ensures that the mean luminance is removed from each training example, rendering the vanilla CNN invariant to global luminance changes, in effect adjusting its sensitivity to global intensity changes. The normalized input then passes through the initial CNN layer, and the output undergoes batch normalization. The subsequent batch normalization layer's trainable and non-trainable parameters are therefore independent of luminance values, and their primary role is to enhance the model's convergence. These parameters do not contribute to altering responses across light levels but rather focus on improving the overall training process. The ability of vanilla CNNs to partially work at different light levels therefore arises from Layer Norm at the input which removes the mean intensity.

We have included this explanation in the Methods (lines 533-553).

— I would appreciate more description of the Angueyra model in the supplemental: at least what the variables represent.

We have included more information about the model in Methods (lines 562-573) and in Supplementary Note. 1.

Reviewer #2

Summary

Idrees et al introduce a convolutional neural network with a photoreceptor model front-end that captures retinal adaptation across light levels. They fit this network to multielectrode-array data from isolated macaque and rat retinas under white noise stimulation at different light levels. The paper focuses on comparing model performance of networks with or without photoreceptor biophysics to conclude that the inclusion of an adaptive front-end generalizes well across light levels. One key highlight (introduced rather late in the paper) is that models of rat retinas trained at photopic light levels can predict well at scotopic conditions. Although this work intersects the domains of retinal neuroscience and computer vision, it offers only incremental advancements in both. While the premise of the study is interesting, the authors never go beyond a model implementation to show how photoreceptor adaptation can contribute to either retinal or computer vision.

Recent observations from the Rieke lab (Anguera et al 2022 and Yu et al 2022) showcase that cone adaptation is relevant for natural vision at timescales of single fixations and how this adaptation can lead to downstream (nonlinear) effects in the retinal output. The choice of white noise stimulation by the authors washes out all the nuance previously shown with natural visual inputs. White-noise stimuli could potentially be analyzed to reach similar conclusions with methods simpler than fitting CNNs. The authors' choice to focus on white noise is particularly unfortunate because it makes the fitted models hard to interrogate further. For example, it has been shown that models trained on natural scenes but not noise can reproduce a wide variety of known phenomena observed with artificial stimuli (Maheswaranathan et al 2019).

The authors' adaptive photoreceptor network is compared with simple CNNs (with BatchNorm) that interpolate photoreceptor dynamics when trained on two extreme light levels (Figure 3). This finding overshadows the impact of this study in the field of computer vision because it reduces the prediction problem in providing the right training data. As the authors also note (lines 300-312), the photoreceptor layer could be replaced by an alternative dynamic layer and modern vision architectures (e.g., with attention mechanisms) offer such highly adaptive solutions. Focusing more on what the biophysical photoreceptor layer offers relative to a generic adaptive layer is a key question to answer. For example, having to fit only four photoreceptor parameters instead of hundreds of generic ones can lead to better generalization with less data.

Overall, the message/conclusion of the study is plausible, but all claims made are only loosely supported by the data. Key methodological details are missing from the text, and the (limited) data are only poorly explored. This data limitation could be overcome with stronger conceptual developments or applications that the authors suggest (lines 92-96).

We agree with the reviewer's assessment and their detailed comments have helped enhance this manuscript. To address the concerns raised, we gathered new primate retina data,

encompassing RGC responses to naturalistic movie stimuli at a finer spatial scale, and used those data in new modeling experiments (new Figs. 2, 3, Supplementary Fig. 1).

Classical CNNs are not designed to extrapolate and therefore it's no surprise that they perform poorly at the task when the testing light level has to be extrapolated. This is a major drawback of classical CNNs in the domain of neuroscience where training data is relatively expensive to acquire. Therefore, one of the claims we make here is that the photoreceptor layer can help the CNN to better extrapolate to different light intensities as compared to the conventional CNN.

To demonstrate that the PR-CNN model can outperform conventional CNNs even at tasks that do not require extrapolation, we now include modeling experiments using data from a new primate retina exposed to naturalistic movie stimuli. We trained and tested the models' abilities to predict these RGC responses at a single light level. The PR-CNN model outperforms the classical CNN model in predicting responses to naturalistic movies even when all models are trained and tested at the same light level (new Figs. 2, 3, Supplementary Fig. 1). This result could not be seen in our previous data with white noise stimuli (Fig 4a,b). We attribute this difference to the temporal correlations and diversity of contrasts present in naturalistic movies that would engage dynamic photoreceptor adaptation mechanisms at a steady-state ambient light level. We believe this represents a substantial improvement in our paper, and we thank all of the reviewers for prompting us to include this result.

What does the biophysical photoreceptor layer offer over other models that may incorporate adaptation? To answer this question, we performed additional modeling experiments on the new primate retina data. In one experiment, we replaced the biophysical PR model with a linear PR model at the CNN's front end. This model's performance was lower than the PR-CNN with the biophysical PR model and was similar to the conventional CNN (new Fig. 3b). This experiment suggests that nonlinearities in photoreceptor processing are a key component impacting retinal responses to stimulation.

Moreover, and highlighting that it is the photoreceptor computation itself that matters – and not just some other dynamical computations that could be captured by the trainable PR model – we performed a second experiment in which we held the PR model's parameters fixed to values obtained from patch clamp recordings from primate rods (performed in Fred Rieke's lab, Chen et al., 2024). This model achieved very similar performance to the one with the trainable PR model, emphasizing that the performance gains can be attributed to the photoreceptor modeling rather than the additional expressivity obtained by including a more complicated input layer.

Finally, and regarding the performance gains from the PR model in our original submission: while they may appear to be somewhat modest, we did observe a substantial performance increase when testing at a light level lower than those at which the model was trained: there, the performance increased from 24% FEV in the conventional CNN to 54% FEV with the PR layer included (Fig. 5c, formerly Fig. 3d). In this case, adding the PR layer doubled the CNN's performance. In our new experiments with naturalistic movie stimuli, the model with the PR layer obtained ~29% higher performance than the vanilla CNN, even when trained and tested at the same light level. Once again, we thank the reviewers for prompting us to pursue the naturalistic stimulus study.

Major comments:

- Experimental considerations

1. The major part of this study's results comes from a single macaque retina recording. Data quality comes to question given that only 37 good cells came out of a dense 512-electrode array recording, which previously other labs showcased that can produce at least hundreds of simultaneously recorded cells (e.g., Rhoades et al 2019). Light responses in the macaque retina are extremely sensitive to preparation conditions; there are empirical standards to assess light sensitivity of macaque retinal tissue (e.g., Rieke, Manookin lab papers). Can the authors provide evidence that the retina was in good condition? How well did cells respond across trials of the test (fixed) white-noise stimulus? Finally, I assume that the retinal piece the authors are analyzing comes from the peripheral macaque retina, but this detail is missing.

We have now detailed this information in Methods. We have also included data from additional primate retina experiments from which we use 57 recorded RGCs in our analysis. Briefly, the RGCs in all experiments were selected based on their spike sorting quality and by our ability to track them across recorded experimental conditions (lines 419-427). Changing light levels can have an impact on spike sorting quality and therefore not all recorded RGCs could be reliably sorted across the 3 light levels. The tissue was in good condition given that it reliably responded across trials. We have quantified the retinal reliability by calculating the variability in responses across trials (lines 428-441). This retinal piece was from the peripheral macaque retina. We have now added this information (line 410).

2. Since the generalization from photopic to scotopic levels is impressive (Figure 7), the study could strengthen its most significant result by analyzing previously recorded data. For example, the Field et al 2009 study, whose first author is also part of this manuscript, includes photopic and scotopic white noise measurements in populations of macaque RGCs. Would the rat results generalize in the macaque condition?

We have included new experiments with primate retina in the updated manuscript showing further interesting results: the photoreceptor model shows performance gains even at a single light level when predicting responses to naturalistic movie stimuli (new Figs. 2, 3, Supplementary Fig. 1).

Given these compelling new results, we therefore focused our manuscript on showing performance gains using a single photoreceptor type (on primate data) as opposed to switching between cone and rod photoreceptor models (like in rat retina experiments).

While the performance gains may appear modest, we believe that they are significant given that across light levels the PR layer doubles the performance of conventional CNNs (Fig. 3c; from 24% FEV to 54% FEV), and when modeling responses to naturalistic movie stimuli at a single light level, the PR layer can improve the performance of conventional CNNs by ~29% for natural movies (new Fig. 3a).

3. Given the limited data, a thorough exploration of the macaque dataset could strengthen the impact of the paper. It is clear from multiple plots that the reported performance is highly variable across cells. Is this variability somehow related to the recorded cell types (midget/parasol)?

We now show performance for individual RGCs. Please refer to Fig. 3, Supplementary Fig. 1 for naturalistic movies experiments at a single light level and Supplementary Fig. 2 for experiments spanning light levels using white noise stimuli. We grouped midget and parasol cells together because of the uneven distribution between the two cell types in our dataset: there were not enough of each cell type for us to break down performance by cell type.

- Model architecture and baseline comparisons

4. While the study is focused on biophysical properties of the photoreceptor layer, the remaining architecture is quite far from the actual retinal biophysics. How was the architecture chosen? The authors state a grid search, but what was the objective and which data were used? Starting from white-noise stimulation, the stixel size used was rather large (140 μm for macaque and 252 μm for rat). At this size length, one pixel is at least as large as a midget ganglion cell receptive field center, and probably consists of multiple bipolar cells. Thus, a single convolutional layer may potentially capture all the operations performed by the three-layered architecture the authors used. The stride used for the first CNN layer (9 pixels) requires integration of information >1 mm which is rather large if we were to interpret the first layer as the photoreceptor or bipolar cell mosaic. Such long-range information integration only happens for wide-field amacrine cells.

The revised manuscript includes new macaque retina data (new Figs. 2, 3, Supplementary Fig. 1) where the pixel size is 30 μm . For all the models in the manuscript, we used an architecture similar to Deep Retina (McIntosh et al. 2016): we chose this architecture as our starting point because it is the state-of-the-art deep learning based retina model. From this starting point, we tuned the hyperparameters using a grid search to determine the number of filters in each layer and filter sizes in each layer. For this hyperparameter search, we used the checkerboard data at the single light level in our new experiments (Figs. 2, 3, Supplementary Fig. 1) and we used data collected at the brightest light level for our previous experiments that spanned multiple light levels (Figs. 4,5). For those previous macaque data, the filter size of 9 and multiple filters in the first layer allowed the model to be flexible enough to cover both short and long range interactions. Indeed the performance gain over using a 2-layered network was not substantial. Nonetheless we wanted to build into the model enough capacity to make sure it was not constrained by size.

5. Although the paper frames CNNs in the center, a multi-layered CNN architecture is not a prerequisite to train models for RGCs in parallel (e.g., Pillow et al 2008). An alternative

that could still reveal photoreceptor effects is to train multiple GLMs (one per cell), all receiving shared input from the adaptive photoreceptor layer. It is possible that the authors can fit such individual models with even less parameters. For example, using (per cell) a 5x5 spatial filter, 10 temporal parameters (using basis functions) plus 2-3 parameters for the output nonlinearity results in 1400-1500 parameters for all 37 cells. It is clear that the CNN approach may be more data-efficient for larger cell counts, but does weight-sharing in the authors' multi-layered CNN architecture actually reduce this individual model parameter estimate by a lot?

It is possible that PR-GLMs will perform similarly to PR-CNN for this dataset, and we include that point in our Discussion, lines 382-388. At the same time, our focus is on the development of the hybrid biophysical-CNN models. Our larger aim – beyond this specific paper – is to build up biologically identifiable models of visual computation that span from retina to cortex. For that work, we are building fully-trainable biologically-detailed CNN layers that can be combined to model retinal (and later cortical) computation. The benefit of this approach is that, at each stage of model development, the regular CNN layers can fill in the gaps between the well-understood biologically-detailed layers, enabling us to still achieve reasonably good prediction accuracy. As a specific example of upcoming work, we aim to replace intermediate CNN layers with trainable bipolar cell models, and to replace the output layer with RGC models that incorporate light dependent spike frequency adaptation. These are major components of a collaborative R01 proposal that is currently under review, and which aims to build on the study presented in the current paper.

- Model comparison and statistical methods

6. The main model evaluation method, fraction of explainable variance (FEV), is actually left unexplained throughout the whole manuscript (despite a reference in line 127). Is this an R-squared-based measure compared to trial-to-trial variability? In any case, the calculation should be reported in the methods. Are cells with negative FEV cells with extremely bad predictions or cells for which the firing was unreliable throughout the recording? Were there any cell inclusion criteria for fitting the model as there were for follow-up analyses (Figure 5)? Did cells show consistent responses over the course of the experiment?

We apologize for the oversight in not including an explanation of FEV in the Methods section. We have now included all this information. Please see lines 624-642. The 37 RGCs pre-selected had reliable firing rates across all the light levels. There was no other inclusion criteria. The cells with negative FEVs are the ones with extremely bad predictions. Cells showed consistent responses over the course of the experiment as trial-to-trial variability was quite low, quantified by retinal reliability (now included in Methods lines 428-441).

7. How well do ganglion cell responses themselves predict responses at different light levels? If one computes FEV for 3R* using the 30R* responses, how much does one

expect to predict? If this number is quite large, I would argue that it becomes a challenge to evaluate whether the photoreceptor layer adds much using these data. This is because there may be little differences in the test set to begin with. This baseline analysis is of particular interest in the case of Figure 7, where the two models compared have drastically different predictions.

Calculating FEV in predicting RGC responses at 30R* using RGC responses at 3R* or 0.3R* yields median FEV values of 73% and 16%, respectively. In our dataset, responses at 30R* and 3R* are fairly similar, but the responses are much more different at 0.3R* (inset in Fig. 4b). One reason for this is that at 0.3R* rod adaptation is fairly minimal: it is thus expected that responses are quite different at 0.3R* vs at light levels above 1R*, where adaptation plays a substantial role.

Notably, this highlights that the most difficult of the generalization between light levels tasks is the one in which the models are trained at 30R* and 3R*, and then tested at 0.3R* (at which the retina is in quite a different adaptation state). In this most-challenging setting, the PR-CNN model shows almost double the performance (54% FEV) as compared to conventional CNNs (24% FEV).

In the rat analysis (Fig. 7), calculating FEV for photopic light level from responses at scotopic light level yields FEV of -82%. This very low and negative FEV is expected as the recorded responses across the two light levels differ significantly. This can be seen in the overlaid recorded responses in Fig. 7a inset.

8. The most concerning observation is that the authors used a sample t-test to obtain their main result (Figure 3), in contrast to the rest of the paper where they use appropriately use non-parametric statistics. Do signed-rank Wilcoxon tests reveal statistical significance between model performance?

We agree. Thanks for pointing this out. We have now replaced that t-test with a signed-rank Wilcoxon test. The statistical significance result is qualitatively similar for these two tests. Below are p values for Fig. 5c (formerly Fig. 3d) comparing the p-values obtained by the Wilcoxon signed rank test and the ttest. The values in brackets represent the p-values obtained from the ttests.

Testing light level of 0.3R*: 4.99e-07 (0.009)

30R*: 1.28e-07 (0.007)

3R*: 0.83 (0.74)

Minor comments:

- The Anguerya et al. study (from which the photoreceptor model comes from) is now published in the Journal of Neuroscience (2022). The authors should update the citation details.

We have now updated this.

- It is unclear whether the length of movie segments used for training is 120 (line 120) or 180 (line 442)? Such a length is a bit unusual for describing temporal filters of ganglion cells. 120 frames at 8 ms each is 960 ms, but primate RGC filters require less than half of that length to capture light responses (this is also seen in the authors' data in Figure 4). Is this long duration to allow for the photoreceptor model to integrate over the recent intensities? A long filter requires more data to fit and could potentially lead to overfitting of temporal components for the baseline CNN model (as seen in Figure 4b).

The length of the movie segments for PR-CNN models is 180 frames. The output of the PR layer is however truncated by 60 frames to get rid of any boundary effects. Temporal filters of 120 frames are indeed long but this also builds enough capacity in the model to account for longer time scale adaptations. Notably, the necessary temporal filter length depends somewhat on light level: we initially chose shorter filters but the performance was low for the lowest light level of $0.3R^*$. In our new experiments of Figs. 2, 3, we have used temporal filters of 80 frame duration, which seems sufficient at light level of $50R^*$ and is also necessary because of limited training data. With this new data, we experimented with different frame durations (60, 80, 120) for the baseline CNN (not included in the paper). Performance on the validation set saturated at 80 frame durations. We therefore set this as the frame duration for both baseline CNNs and PR-CNNs.

- Figures 2,6 and 8. Are the RGC responses shown averages of repeated noise segments or smoothed single-trial responses? Because they mention repeated segments of five to ten seconds long, it would be informative to compare model responses across the whole segment and not just two seconds.

The responses shown are averages of repeated white noise segments. For clarity, the figures show only 2s of data. For model performance calculations (FEV), the entire segments were used. The duration of repeating validation segments (held out from training) used for quantifying performance in the new primate data is 6 seconds. We have now explicitly mentioned this in line 106

- How were rod isomerization values calculated?

We measured the spectral intensity profile $\mu W cm^{-2} nm^{-1}$ of our light stimuli with a calibrated spectrometer. We transformed the stimulus intensity into equivalents of photoisomerizations per receptor per second. The spectrum was converted to photons $cm^{-2} s^{-1} nm^{-1}$, convolved with the normalized spectrum of macaque cones and rods, and multiplied with the effective collection area of these photoreceptors. We have now included this information in lines 446-452 and 480-487.

- How do the estimated photocurrents differ for scotopic and photopic levels in the rat retina? The differences should be more dramatic compared to the analysis of Figure 6f.

The differences are much more dramatic as the photocurrents are much more smooth at scotopic than at photopic levels. This makes it difficult to analyze latency changes simply from photocurrent.

- Are the rat retina results symmetric over the two recordings? If trained on Retina B, the photoreceptor model should also generalize to Retina A.

We have not explicitly tested this. In the revised manuscript, we focused more on describing the potential of the PR model using a single photoreceptor type – namely rods. Importantly, our new results emphasize that, even at a single light level, the PR model leads to performance gains in predicting responses to naturalistic movie stimuli, as it can capture nonlinear local adaptation under dynamic luminance conditions.

In our future work, we will return to the important question of generalization across very different light levels. In that work, we intend to train models with multiple channels of photoreceptor input (e.g., a rod channel and 3 cone channels) and to train on data from photopic, scotopic, and mesopic light levels. The hope is that the resultant model will automatically capture the switch from rod-dominated to cone-dominated signal processing, as the light level increases, without us having to manually switch the PR model type the same way we did here for this (somewhat more preliminary) investigation.

- Units of firing rates in single-cell examples. In Figure 2, 6 and 7, it is unclear whether the firing rate is normalized or not. A normalized measure typically has no unit (in the Figures, instead, spikes/s is given). For these plots, it would be helpful to examine the actual response magnitude (in spikes/s) and also where the zero baseline lies.

The models were trained on normalized spike rates. We normalized the spike rate of each RGC by its median spike rate. This was to ensure that firing rates of all units are on the same scale so that the model output unit weights also learn at the same scale. We have updated the figures to remove the units.

- The authors should explain how they decompose receptive fields into spatial and temporal components. Do they just use the stixel frame/timecourse where the highest absolute value occurs?

We factorize the spatio-temporal RF matrix through Singular Value Decomposition to obtain separate spatial and temporal components. We have now mentioned this in Methods (line 652).

- The rat recordings contained only OFF cells. Is there a reason that ON cells were not included?

The rat datasets were part of a previous study (Ruda et al., 2020). ON RGCs were recorded but not included as part of the datasets used in that study as their mosaic was incomplete, which was important for modeling response correlations in that study. We used the same dataset that excluded ON RGCs for our models. We don't expect significant differences in our results with ON RGCs included given that for primate experiments, we observed similar performance across ON and OFF cells (Fig. 3 and Supplementary Figure 2).

Reviewer #3

The paper discusses a novel convolutional neural network (CNN) for vision that integrates a biophysical model of the phototransduction cascade within the photoreceptors. This novel model, referred to as the photoreceptor-CNN (PR-CNN), significantly enhances the prediction of retinal ganglion cell (RGC) responses across varying light levels, outperforming conventional CNN models. I believe this paper represents an important advancement in artificial neural network (ANN) models of the visual system. While the primary focus of the paper lies in demonstrating the benefits of incorporating a realistic photoreceptor layer for predicting RGC responses, its implications extend far beyond, potentially impacting our models of the entire visual system and downstream higher-order visual areas. This paper is a great contribution to both neuroscience and AI, and I support its publication.

However, I have a few comments that need to be addressed to enhance the clarity of the paper:

Major comments:

1- Although the advantages of the bio-realistic photoreceptor layer in predicting RGC responses are evident, as a reader, I found it challenging to grasp the mechanistic model of photoreceptors. While the authors do provide the equations of the photoreceptor in the appendix, an introduction to the model and some intuitive explanation within the main text would greatly help.

We agree and have now included an explanation of the model in the main text (lines 113-121). We have also included a schematic of the phototransduction cascade that the model captures (Fig. 1c).

2- Considering that the photoreceptor model involves dynamic nonlinearities, it would be beneficial to explore whether increasing the depth of the regular CNN model leads to better predictions of RGC responses. Is there a specific dynamical motif that multiple (recurrent) layers might fail to learn and represent? Is the improvement in performance attributable to the larger capacity of the model with the photoreceptor layer or the specific inductive biases of the photoreceptor layer itself? Conducting more detailed experimentation, such as ablations, involving different trainable components of the photoreceptor layer could provide valuable insights, particularly in understanding the computations required for context-dependent visual processing. Addressing this concern is closely related to the need for an intuitive understanding of the photoreceptor layer to comprehend the underlying reasons behind the obtained results.

These are very important suggestions and we have incorporated them in our revised manuscript. Specifically, we now compare performance for models where (1) PR layer parameters were trained through backpropagation, (2) PR layer parameters were fixed to their biological values, (3) the biophysical PR model was replaced by a linear PR empirical model.

From these experiments we observed that: (a) the PR-CNN with biophysical PR model outperforms the rest (Fig. 3, Supplementary Fig. 1); (b) the performance gains are not substantially changed by setting the PR layer to be trainable vs using the parameters determined through experimental fits to patch clamp recordings from rods (through previous experiments in Fred Rieke's lab) (Fig. 3c); and (c) the performance of the PR-CNN with the linear PR model is similar to that of conventional CNNs (Fig. 3b). In summary, nonlinear adaptive mechanisms and calcium feedback in the biophysical PR model appear to be the key to increasing model performance. It is worth noting that the photoreceptor layer only adds an additional 12 parameters, out of which only 5 are trained in the trainable version: this is quite a modest increase relative to the 873,642 trainable parameters in the rest of the CNN model. In fact, with the PR model as a front end, the downstream CNN can be further optimized, reducing the trainable parameters of the rest of the CNN to ~538,107.

These new results are included in the first two sections of the revised manuscript.

3- The performance of PR-CNN in the $30 R^* \text{receptor}^{-1} \text{s}^{-1}$ condition appears to be lower than that of the CNN model. It would be beneficial if the authors could provide an explanation for this discrepancy and determine whether it indicates a systematic change.

While the median FEV of the PR-CNN model is 2% lower than the CNN model, the error bars are much smaller. This is not a systematic change as can be seen by a detailed comparison of FEVs for each RGC (Supplementary Figure 2).

Minor comments:

1- Figure 5 requires color bars to distinguish different bins clearly.

This figure is now Supplementary Figure 4. We have now included the color bar in panel b.

2- In Figure 4, specifically in panels b and c of the bottom row, the use of different colors needs clarification.

The Figure is now Supplementary Figure 3. The different colors are consistent with the different colors used in Figs. 5c, and Supplementary Figure 4d-f (formerly Figs. 3, 5d-f) to differentiate between the conventional CNN and photoreceptor CNN model.

3- Line 236 mentions, "This change in kinetics could already be observed at the output of the photoreceptor layer." It is not entirely clear what changes in kinetics in Figure 6f should be observed. The authors should clarify this point for better understanding.

The change in kinetics in this case refers to the slowness of photoreceptor output at the lower light level. We have now made this more explicit (line 278).

Reviewer #4

Summary of the manuscript:

The authors present a novel convolutional layer to add as a first entry layer before traditional convolutional neural networks models (CNNs) of the retina.

Their main claims are that this layer, called Photoreceptor Convolutional Layer, which is based on a biophysical model previously published, are:

- 1- that the new model can correctly capture changes in sensitivity, which they call "gain".
- 2- that the new model captures temporal dynamics that change according to the intensities of the input images
- 3- that these properties allow the model to generalize to different ranges of intensity, outperforming traditional CNNs

They perform modeling on retinal ganglion cells (RGC) on a new dataset from one monkey retina and in two datasets already published from two rats retina.

The research is novel and pertinent. If proven generally successful, it may be included as a first layer in artificial intelligence (AI) vision processing models. This will be a good advancement in the field of research interested in obtaining biologically plausible and realistic models, in order to gain insight from the parameters and inner layers of models, in contrast to the more general "black box" approach that deep neural networks in AI modeling usually have.

However, I think the authors could have done a much better job to prove the advantages, and specially to characterize the properties of their new neural network with a biophysically inspired input layer.

We thank the reviewer for their detailed comments, which we have addressed in a major revision of this manuscript. Specifically, we performed new modeling experiments with new primate retina data in which the retinal was stimulated with both naturalistic movies and with white noise stimuli. Applying our model to these data, we find that (1) even at a single light level, the PR model increases the performance of vanilla CNNs by ~29% (in terms of FEV) in predicting responses to naturalistic stimuli (new Fig. 3a). Our previous results showed that across light levels, the PR model can lead to performance gains by up to 125% (i.e. increase in performance from 24% with conventional CNN to 54% with the photoreceptor layer included; Fig. 5c). We believe that these are substantial improvements, although there is still more work to be done as the models are not yet near the ceiling set by perfect performance.

To characterize the properties of the PR-CNN model, we performed variants of our modeling experiments in which: (1) PR layer parameters were fixed to their biological values, and (2) the biophysical PR model was replaced by a linear PR model. From these experiments we observed that: (a) the PR-CNN with biophysical PR model outperforms the rest (Fig. 3); (b) the performance gains are not substantially changed by setting the PR layer to be trainable vs using the parameters determined through experimental fits to patch clamp recordings from rods

(through experiments in Fred Rieke's lab published recently in Chen et al., 2024) (Fig. 3c); (c) the performance of the PR-CNN with the linear PR model is similar to that of vanilla CNNs (Fig. 3b). In summary, nonlinear adaptive mechanisms including the calcium feedback in the biophysical PR model appear to be the key to increasing model performance.

These new results are now included in the first two sections of the revised manuscript.

Major comments:

1) INTERPOLATION VS EXTRAPOLATION: My first immediate question when reading the manuscript was, why do the authors train in the upper and middle range of intensities and test in the lower, instead of training upper and lower ones and testing in the middle? I was expecting, as is usual for AI models, that the standard CNN they were trying would not extrapolate to "out of training range" intensity values, as the authors show. However, they later show that they trained the model as I expected, and that it performs as good as the new model they propose. Then, all the rest of the article is focused on the capability of the model to generalize when extrapolating, and comparing its performance with the classical CNN. This is somehow killing their argument that their model is better "per se". In my view, is that they just trained the classical CNN with the wrong dataset.

However, if their claim was focused on that the Photoreceptor layer can extrapolate to different intensity ranges, this is a different story. The authors should make this clear and more transparent.

We agree that classical CNNs are not designed to extrapolate and therefore it is no surprise that they perform poorly at this task. This is a major drawback of classical CNNs especially in the domain of neuroscience, where training (and testing) data is fairly expensive to obtain. Therefore, one of the claims we make here is that the photoreceptor layer can extrapolate much better to different light intensities as compared to the conventional CNN. At the same time, to demonstrate that the PR-CNN model can outperform the CNN model in situations that do not require extrapolation, we now include modeling experiments using data from a new macaque retina that was stimulated with naturalistic stimuli and where we trained and tested the models at a single light level. In these experiments with naturalistic stimuli, the PR-CNN model outperforms the classical CNN model even when trained and tested at the same light level (new Figs. 2, 3). We attribute this effect to the temporal correlations and diversity of contrast values present in naturalistic scene movies, which may activate dynamic photoreceptor adaptation even at a single light level. This effect could not be seen in our previous experiments with binary white noise stimuli, which have an uncorrelated stimulus structure and less diversity of contrast values.

2) IS CNN INTRINSICALLY WORSE?: The performance of the models when interpolating is similar (Figure 3d). Then, it would be interesting to compare the parameter convergence of the models to understand what is failing for the CNN when extrapolating, or what characteristics of the photoreceptor layer is modeled into the CNN that works better (since the best PR-CNN is comparable to the CNN). Is the CNN of figure 3d able to capture the claims for the PR-CNN

shown in Figures 5 and 6? If the answer is yes, then having a PR-CNN does not imply that this type of model can "only" capture points 1- and 2- in the summary above. The authors should explicitly compare the number of parameters of each model. Is it that the PR-CNN is better because it has more parameters? I doubt this, but a clarification can help. Related, how would Figure 6d look for the interpolated CNN. Can this CNN capture time kinetics even without a PR-CNN?

We limited the gradient analysis of Figs. 5 and 6 (now Fig. 6 and Supplementary Fig. 4) to the training light levels since the objective here was to show that even at training light levels, the photoreceptor model can capture the different kinetics associated with different light levels. We have now included details on the number of parameters for each model. For the new primate data we model, conventional CNN has ~800k parameters (line 88) whereas PR-CNN has ~500k parameters (see line 154). This difference in parameter count resulted from separately optimizing the hyperparameters of the photoreceptor-CNN and conventional CNN models via grid searches.

Concerning Fig. 6d, classical CNNs have no inherent knowledge of light levels and therefore are unable to exhibit different temporal RFs for different light levels. We did not explicitly test this for the interpolation condition. Instead, we included new experiments in which a primate retina was stimulated with both white noise and natural scene stimuli, and we subsequently trained and tested the CNN models using data from a single mean light level. These new experiments show that the photoreceptor-CNN model outperforms the classical CNN model even at a single light level, suggesting that the benefits of the photoreceptor-CNN model extend beyond the cases of interpolation to new light levels.

3) WHAT DID WE LEARN?: The performance in general is poor at this stage of the model when extrapolating (around 50%), compared to the more than 80% achieved when training and test are the same. So, apart from the fact that a photoreceptor layer can extrapolate to other intensity ranges, is there a difference between the three models learned when testing at different values? As I understand, they retrained each time. So, can we compare the parameters obtained for the PR layer (they are only 4, and they could maybe easily studied/interpreted). Related, in line 324 authors say: "the biophysical photoreceptor model we use in the photoreceptor layer while complex, has parameters that map directly onto the biology, providing an opportunity for investigating the role of photoreceptor adaptation in the retina." (plus following sentences). Can the authors show/interpret this with their fitted parameter models?

We have now included the initial parameter values in Supplementary Table 1. In most cases we set the photoreceptor model to be trainable. In this setting, the learnt parameters may not only capture adaptation in the photoreceptor but also some parts of downstream adaptation that can be captured by adjusting the parameters of the PR model. In some of our new modeling experiments, we fixed the photoreceptor model parameters to their biological values from fits to rod data obtained as part of other experiments on a different macaque retina. The model with fixed PR values performed similarly to the one where the parameters were allowed to be trained

(new Fig. 3c). Moreover, this new model variant maps the parameters directly onto the biology and will be useful in studying how receptive fields depend on such adaptation. We leave this study for future work.

One way to investigate the role of photoreceptor adaptation is to knock out certain components of the PR model. We demonstrate this by performing a new modeling experiment (line 150) where we replaced the nonlinear photoreceptor model with a linear photoreceptor model (new Fig. 3b). This model did not perform as well as the biophysical model, which incorporates nonlinear adaptive mechanisms as opposed to linear adaptive mechanisms in the simpler model. This suggests that nonlinearities in the photoreceptor model, and feedback mechanisms, are important factors underlying the observed performance gains.

4) RODS vs CONES. The rationale to go from rods to cones is not smooth. First, when they introduce the light levels, they do not motivate it. Why would they want to study rod dynamics and not cone dynamics? Is it because it will be easier to treat only one active photoreceptor and capture its dynamics? Then, the reference the authors mention, where the PR model is built (Angueyra et al., 2021), presents data ONLY for cones (for monkeys in another intensity regime). Then, wouldn't it be necessary to make at least a comment on this transition? Is it reasonable to use a model fit for cones directly for rods? Should this be later checked experimentally? Regarding the methods, what is the rationale for fixing 8 parameters?, what is the bibliographic reference to find these fixed values? I understand the code will be available, but a table with the values chosen may help. More so if there are two different types of photoreceptors fitted in the manuscript. Is there an intrinsic difference between them? If the model of Angueyra et al., 2021 (still in preprint form) is the first to describe this, then in this manuscript, more effort could be put to show the difference between cones and rods in the model output and/or parameters. I understand that the authors want to put the equations in supplementary material, but some diagram like Figure in Fig6A of Angueyra et al., 2021 will be of much help.

Finally, do Figure 5 and 6 results hold for photopic levels (or maybe show more drastic differences?). Or cones and rods present different dynamics for this data? (also monkey and rat data may differ). I see in Fig7a that the fit is bad. (maybe too bad), so maybe this quantification cannot be done? Could the authors confirm that this is the best possible fit for CNN without a photoreceptor layer? Or is it that I am misinterpreting Fig 7a (it may be because this section is a bit disorganized and difficult to follow). I think Fig7 result will be very much more impactful, if all my comments above are addressed. I had to make a lot of effort to understand what was presented.

We appreciate this comment and note that there is a key point that we did not sufficiently explain in our original submission: namely, that the same biophysical signaling cascade model (albeit with different parameters) applies to both rod and cone photoreceptors. The Angueyra et. al paper focuses on cones, but that same model has been fit to recordings from primate rod cells (Chen et al., 2024), using the same procedure as in Angueyra et al. Among the studies included in our revised paper are ones in which the PR model parameters are held fixed at their experimentally-derived values (derived from patch clamp recordings from primate rod cells,

Chen et al., 2024) (Fig. 3c). We have clarified this point in the revised paper, on lines 119-121 and we have listed the parameter values for both rods and cones in Supplementary Table 1.

In the revised manuscript, we have focused more on describing the potential of the PR model using a single photoreceptor type – namely rods – and how the model leads to performance gains using natural movies at a single light level. Nonetheless, we fully agree with the reviewers assessment that this section was disorganized and lacked information that could have improved it. We have now included additional figure panels (Fig. 7) and also moved the training schematic from Supplementary to the main manuscript (Fig. 7f).

Briefly, the rationale for starting with one photoreceptor type was that we wanted to capture dynamics of just one photoreceptor type. We chose rod light levels because at very low light levels, adaptation is minimal. This would therefore allow one to investigate the role that rod adaptation plays in overall retinal signaling (not part of this study. The data was collected with the intention of using it for multiple interlinked studies).

We have also included a photoreceptor model figure as suggested (Fig. 1c). Cones and rods will present different dynamics since rod responses are much slower. Therefore it will not be so straightforward to make the quantification of Fig. 5 and 6 with the rat data across extreme light levels.

Fig. 7b (formerly Fig. 7a) is the best fit for the CNN without the photoreceptor layer: the hyperparameters of that CNN model were optimized using a grid search.

5) SINGLE CELL DATA: Although the main claim of the model performing better is in Figure 3 showed as percentages in the population with box plots and outliers, a figure with the individual performances comparison for individual cells is missing and is a common practice to assess the goodness of a model (see outliers, general trends, etc.). Other measures may be used also, to see if the PR-CNN model is doing better for just a subset of cells, or if it makes it better for all of them.

We have now included pairwise comparisons for each RGC (Fig. 3 for new experiments and Supplementary Fig. 2 for previous experiments across light levels).

Minor comments:

1) The authors mention "changing input conditions", "sensitivity", "gain", but are vague in general, until getting to the results. It may help to be more specific onto which type of changes they refer to, and use a single nomenclature across the manuscript.

Agree. We have now made the wordings more specific.

2) The way data is first presented is confusing. The percentages in the main text do not match the percentages one first encounters in Figure 2. It does not help that one of the population % matches the example given (78%). This can be easily changed by reordering figures or text.

We agree, and we have adjusted this in our revision. Please see lines 213-216.

3) The "R* receptor-1s-1" is cumbersome to read each time the authors mention the light levels, and could be easily replaced by R^*/μ or directly R^* if defined the first time it appears. It is cumbersome to read it. It could also be avoided by naming H M and L the high, medium and low levels of stimulation. Same for the Layer Normalization Layer: could be said Normalization Layer.

We have now used the terminology high, medium and low light levels (lines 194-195). Neural networks can have several different types of normalization layers such as Layer, Batch, Instance Normalization. Layer Normalization is a specific type and clarifying which type of normalization is being used may be important for other researchers interested in building on our methods. They can of course use the code that we share, but if they are including some of our approach into their own models, they may benefit from us being explicit about the specific type(s) of normalization we included.

4) Maybe I am wrong, but it seems that here there is an extra layer compared to Deep Retina (the 18 channel one). Is this correct? Was this necessary here for the Photoreceptor layer to work better? If this layer is not present, PR-CNN does not work, or is it just worse?

There is an extra layer but it also exists in our vanilla CNN. The network architecture of the vanilla CNN is based on Deep Retina, but with all of its hyperparameters – including the number of layers – tuned to our dataset. This tuning included adding an additional layer to make the network deeper. The PR-CNN would work even if this layer were removed but we wanted to make sure that the performance we report for the conventional CNN is as high as possible, to make the fairest comparison with the PR-CNN.

5) "Normalization by mean and variance" sounds strange, you mean z-scoring? In line 120 says it has a similar effect to z-scoring, then I do not understand the difference.

We have clarified the explanation. We wanted to be specific about the normalization axis. Revised explanation is in lines 93-96).

6) A formula for the FEV will be helpful in the methods. Especially for interpreting the negative values.

We apologize for this oversight and have now included a detailed description and formula for FEV (line 624).

7) Line 138 "it predicted the same output for a given movie segment". I do not see this explicitly proven later in the manuscript, although somehow Fig 6c points on that direction.

We have now removed this sentence to avoid confusion.

8) Line 141, Input conversion, the authors mean just "different input ranges"?

We mean the pixel intensity values. We have however significantly re-worded this section.

9) There is an inconsistency between the frequency reported for the movie (60/30Hz), and the frame length used for the modeling (8ms).

Does this mean that the movie frames were stepped in subframes? How did the authors treat the data?

Stimuli were upsampled to 120 Hz by repeating the frames. Spikecounts were binned in bins of 8 ms. The upsampling was necessary for the PR model in which differential equations are solved using the Euler method.

10) When taking out some data, a plot of the performance of the cells could be helpful, to understand the rationale of the threshold selected (Line 210), since values at around 50% are quite low to be "more reliable".

We wanted to have a decent number of RGCs to be able to perform population analysis. Therefore this was a tradeoff. A detailed performance plot is now included in Supplementary Figure. 2

11) I would expect Figure 4a to be a step function, but I see diagonal lines. In the case of the CNN in 2b, an explanation to understand why is much less smooth than the PR-CNN could be helpful. Finally, the method to decompose space and time is not explicitly specified in the methods.

This is a rendering error. We have now fixed this (now Supplementary Figure. 3a). We decomposed the receptive fields into space and time by using Singular Value Decomposition. This is now included in Methods (line 652).

12) In figure 6a, if a curve is normalized, then shouldn't it be in arbitrary units, and below 1?

We have corrected the units. The normalization is with respect to the median response of an RGC and not with respect to the highest value. We have now included this information. Please see lines 98, 515.

13) the differences in Fig6f do not stand out, I would zoom in or include an inset.

We have now included an inset in Fig. 6f.

14) The fact that the gradients are so smooth in PR-CNN does not come straightforward. This seems an advantage of the model.

This could be an advantage of the model, although the CNNs seem to be doing well at the training light levels.

15) "We leave that analysis for future work". This type of phrase could be ruled out. If makes it feel that this should be done now in this manuscript. Same for "future studies will", "future work will" change it for "may".

Noted. We have modified the text with this suggestion.

16) 10% refractory period contamination seems a cortical like extracellular recording. I think that the standards for MEA in vitro are much better. Some papers reporting less than 1% at 2ms windows. Authors should revise this to explain why such high values are permitted.

10% refractory period contamination has been a standard in our labs and has produced reliable results in the past (Roy et al., 2021; Shlens et al., 2009; Field et al., 2007; Yu et al., 2017). However, to address this concern, we looked at the spike contamination percentages for the primate retina recorded using this setup. Most of the sorted units had 0% spike contamination based on refractory period violations. Other units had contamination in the range of 0.05% to 0.09% with one unit at 0.8%. We have added this information in the Methods section (lines 423-427).

17) Labels in Fig6a is superimposed with numbers.

We have now fixed this.

18) Something like Supplementary Figure 1 (with a better description), may help very well to introduce/explain the photopic vs scotopic experiment if included in Figure 7.

We have now added the schematic to Fig. 7.

REVIEWERS' COMMENTS

Reviewer #1 (Remarks to the Author):

The authors significantly revised the manuscript and added some beautiful additional work that supports (and extends) the conclusions of the paper as well as clarifies many of the details. In particular, the entire section demonstrating the superior performance of the hybrid photoreceptor-CNN model was a strong addition including Figures 1 and 2, and also serves to offer a much clearer explanation of how their results can be explained, and the whole model was motivated. The additions to the Discussion and Methods were also quite useful, and I appreciate the honest assessment of the limitations of their work through the Discussion and also their response to reviewers, which also serves to make clear what their advances are.

I stress that I find this body of work quite significant for reasons described in my last review. Taken with their revisions, they demonstrate convincingly that adding a the biophysical photoreceptor model at the front end of a CNN (and showing that it is trainable) can result in a much more accurate and interpretable model of subsequent visual processing. This work encompasses both the conceptual advance of the model and its use, the right experiments, and many technical advances to make this trainable and have the right components. This is not only important work in its own right, but — as the authors state — sets a foundation for much impactful future work. In particular, establishing a more biophysically grounded front end of visual models will provide a basis for all subsequent models of later stages of visual processing (not just retinal ganglion cells but throughout the visual pathway) more interpretable, since they will not try to capture these photoreceptor nonlinearities in their nonlinear structure but will correctly attribute them to the first stages of processing.

The addition of the new section also brought several smaller problems, which are mostly about methods and clarifying these new experiments. There was a lot of material that was briefly described, and the text as written mixes in a number of unexplained methodological details that distract from the important points being made. I have both more important and smaller comments there, and think that revising this one section would further enhance the impact of the paper.

MORE SIGNIFICANT CLARIFICATIONS (but still small)

1. Overall comments on the first section of results: there were a lot of detailed methods added to the description of the model that has no context and not enough explanation for the general reader, and most of the numbers and many of the details about the layers would ideally be put in the methods, and qualitative descriptions that left. I would suggest the bar of "what would a general reader need to understand to make sense of the results?" for this. Also for the non-general reader, the inclusion of some of the specifics (but not all) raises several unanswered questions that are not addressed before a full explanation in the Methods. Some of these are listed as smaller comments below. In comparison to the methods-like description of CNN, the summary of the photoreceptor layer following it (>line 109) is quite nice, and at a good level of detail.

2. What does "nonlinear adaptation" refer to? I ask this both from the sense of terminology, but that this terminology encompasses the main motivation of their paper. This terminology is important and easily confused: one could technically consider divisive normalization as nonlinear, but this is handled by the LayerNorm and not by the photoreceptor mechanisms at all. What the authors do seem to mean is probably based on the comparison to a model that just has a non-adapting linear kernel (where the kernel has 5 parameters that do not change across the stimulus conditions? So is in either case, the output of the photoreceptor linear, but the model specifies how the effective parameters of the kernel change? Or is the photoreceptor model not capturable by a linear kernel at all on short-time scales? These ambiguities are technically resolved in the Methods where the 5-parameter model is explicit, but the main points are currently obfuscated in the results. I would suggest leaving methodological details out but more clearly describing what "nonlinear adaptation" means and then,

how this is demonstrated with the alternative model.

3. One of the major points, why the parameter count was so much smaller in the photoreceptor layer, could be much better explained, and is only in lines 155-156, using an unexplained "grid search" — what was searched over a grid?. I think what it is saying is that the size of the CNN (and photoreceptor-CNN) were optimized to best predict responses, and the CNN had to be much larger without the photoreceptor layer to optimize responses, and even then it performed worse? The sentence present does not capture this, and is a major point in favor of the results of this paper. Does this imply that fixing the same CNN structure would make the non-hybrid CNN do a lot worse? So, in addition to simply clarifying what is meant here, the explanation could cast the results reported in an even better light.

4. The additional analyses described in the paragraph starting on line 174 are confusing and should be better described/elaborated. This paragraph seems to be drawing conclusions, and then explaining an experiment (again slightly too briefly) that these conclusions are based on, so cause/effect and relationship to previously described results is obscured. I assume this is another test independent test of the model — but what stimulus was being used, and why was adaptation not required? Also, I thought that — once trained — all the models were tested on fixed parameter values. So what were these fixed on? (was it not trained)?

SMALLER EXPLANATIONS/CLARIFICATIONS

— The description of cell criteria was added to the Results (lines 78-9) but "reliability across experimental conditions" was vague and only explained in detail in the methods. It would clarify to simply say that "reliability" here is not a general/ambiguous term but refers to detailed criteria explained in the methods.

— Line 86 about batchnorm: This is now very well described in the methods, but the sentence here in the text gives the wrong impression, I believe. As stated in the methods, this is a running average during training only but that results in fixed input-output for evaluation and prediction purposes, and does not involve dynamic z-scoring like the LayerNorm. As written, this seems to imply that it is a dynamic "batch-by-batch" z-scoring, which should be clarified.

— lines 93-93: the LayerNorm layer: it seems to say that the mean across all time points in each movie (80 frames) was removed on a pixel-by-pixel basis, which seems that it would disrupt/cause spatial contrast where there was not any. Am I understanding this right, and would this be a problem? I believe this is probably fine, but could be better described.

— Line 121: Why were photoreceptors configured for rods for the whole experiment? Is this just for the experiment with the animated natural images? Why was this the case? Was this the one demonstration (additional experiment) that was performed or was it necessary for the model to work at any luminance. Please also specify whether this rod configuration was indeed for all light levels, or just the first section

— Line 131: What parameters about the photoreceptor model were learned (line 132)? Not asking at the level of methods, but what did the learned parameters control: "i.e. time scales, the relative amount of cGAMP pool, etc?): can this be easily summarized?

Reviewer #2 (Remarks to the Author):

The authors have adequately addressed all my comments and substantially improved the manuscript's content. I have no further comments.

Reviewer #3 (Remarks to the Author):

Thanks to the authors for their comprehensive responses to the reviewers' comments. All of my concerns have been addressed in the revised manuscript. I recommend this paper for publication.

Reviewer #4 (Remarks to the Author):

The authors thoroughly addressed all of my comments. My main concerns were fairly well solved by adding a new experiment with natural images dynamically shifted using saccade movements. The new data, figures, organisation and better writing of the manuscript profoundly increased its quality. I believe that this article will provoke the community into questioning the need to include biophysical modeling into CNN architectures for vision. However, much work lies in the future in order for these models to perform better and to add more precise insights into retinal physiology.

Some final minor remarks:

Line 154: it is not clear how a grid search can decrease more than 300,000 trainable parameters. Are the authors focusing on a specific layer parameters?

Figure 3b: is there a difference between ON and OFF cells? It seems that a linear photoreceptor model may be good for many OFF cells.

Figure 2: Since an interesting claim is made about the fixed photoreceptor model CNN (Fig 3c), it may be nice to see an example of its behavior in a Figure 2c.

Line 219: Could the authors make a rationale on why they obtain around 40% more of FEV for white noise model (88%) vs the natural movies (49%)? Do midget cells have a higher FEV, and then this explains the difference? (different balance of cells types are included in each dataset). Is the model proposed not enough to predict as nicely the natural movies? Is there any improvement the authors foresee will point in the direction of achieving better results, as the 88% reported from Figure 4?

Line237: "We attribute this to the model's ability to modulate output properties based on mean light level". This claim, although it seems very plausible, is not properly explained with data. What property are the authors talking about? A detailed list of the fitted parameters of the model is now given, and there is a parameter changing 2 orders of magnitude. Is this one the main responsible for the success of the PR model across light levels? I think this is a missed opportunity, as they say in line 373, that other models could be used to obtain similar results.

Line 470: "This was necessary as we first trained the models on checkerboard movie and then fine-tuned the same model with naturalistic movies". This is a technical reason, but not necessary, they could have upsampled the checkerboard movie.

Responses to reviewer comments on:

“Biophysical neural adaptation mechanisms enable artificial neural networks to capture dynamic retinal computation”

Idrees, Manookin, Rieke, Field and Zylberberg

We thank the reviewers for their insightful comments to help finalize our manuscript. In what follows, we provide specific responses (colored in green text) to the reviewer comments (black text), which we have incorporated into the final version of this manuscript.

REVIEWER COMMENTS

Reviewer #1

The authors significantly revised the manuscript and added some beautiful additional work that supports (and extends) the conclusions of the paper as well as clarifies many of the details. In particular, the entire section demonstrating the superior performance of the hybrid photoreceptor-CNN model was a strong addition including Figures 1 and 2, and also serves to offer a much clearer explanation of how their results can be explained, and the whole model was motivated. The additions to the Discussion and Methods were also quite useful, and I appreciate the honest assessment of the limitations of their work through the Discussion and also their response to reviewers, which also serves to make clear what their advances are.

I stress that I find this body of work quite significant for reasons described in my last review. Taken with their revisions, they demonstrate convincingly that adding a the biophysical photoreceptor model at the front end of a CNN (and showing that it is trainable) can result in a much more accurate and interpretable model of subsequent visual processing. This work encompasses both the conceptual advance of the model and its use, the right experiments, and many technical advances to make this trainable and have the right components. This is not only important work in its own right, but — as the authors state — sets a foundation for much impactful future work. In particular, establishing a more biophysically grounded front end of visual models will provide a basis for all subsequent models of later stages of visual processing (not just retinal ganglion cells but throughout the visual pathway) more interpretable, since they will not try to capture these photoreceptor nonlinearities in their nonlinear structure but will correctly attribute them to the first stages of processing.

The addition of the new section also brought several smaller problems, which are mostly about methods and clarifying these new experiments. There was a lot of material that was briefly described, and the text as written mixes in a number of unexplained methodological details that distract from the important points being made. I have both more important and smaller comments there, and think that revising this one section would further enhance the impact of the paper.

We appreciate these supportive remarks and thank the reviewer for their constructive feedback.

MORE SIGNIFICANT CLARIFICATIONS (but still small)

1. Overall comments on the first section of results: there were a lot of detailed methods added to the description of the model that has no context and not enough explanation for the general reader, and most of the numbers and many of the details about the layers would ideally be put in the methods, and qualitative descriptions that left. I would suggest the bar of "what would a general reader need to understand to make sense of the results?" for this. Also for the non-general reader, the inclusion of some of the specifics (but not all) raises several unanswered questions that are not addressed before a full explanation in the Methods. Some of these are listed as smaller comments below. In comparison to the methods-like description of CNN, the summary of the photoreceptor layer following it (>line 109) is quite nice, and at a good level of detail.

We agree with this assessment and have now included a more intuitive description of the CNN architecture in the main text and moved the specific details to Methods. Please see lines 103-114.

2. What does "nonlinear adaptation" refer to? I ask this both from the sense of terminology, but that this terminology encompasses the main motivation of their paper. This terminology is important and easily confused: one could technically consider divisive normalization as nonlinear, but this is handled by the LayerNorm and not by the photoreceptor mechanisms at all. What the authors do seem to mean is probably based on the comparison to a model that just has a non-adapting linear kernel (where the kernel has 5 parameters that do not change across the stimulus conditions? So is in either case, the output of the photoreceptor linear, but the model specifies how the effective parameters of the kernel change? Or is the photoreceptor model not capturable by a linear kernel at all on short-time scales? These ambiguities are technically resolved in the Methods where the 5-parameter model is explicit, but the main points are currently obfuscated in the results. I would suggest leaving methodological details out but more clearly describing what "nonlinear adaptation" means and then, how this is demonstrated with the alternative model.

We agree that the terminology can be confusing. We therefore now use only the term adaptation and state that this adaptation originates as a result of nonlinearities in the model (lines 234, 281-283).

3. One of the major points, why the parameter count was so much smaller in the photoreceptor layer, could be much better explained, and is only in lines 155-156, using an unexplained "grid search" — what was searched over a grid?. I think what it is saying is that the size of the CNN (and photoreceptor-CNN) were optimized to best predict

responses, and the CNN had to be much larger without the photoreceptor layer to optimize responses, and even then it performed worse? The sentence present does not capture this, and is a major point in favor of the results of this paper. Does this imply that fixing the same CNN structure would make the non-hybrid CNN do a lot worse? So, in addition to simply clarifying what is meant here, the explanation could cast the results reported in an even better light.

Thanks for this suggestion. We agree and have now updated the text in line with these suggestions. Please see lines 240-247.

4. The additional analyses described in the paragraph starting on line 174 are confusing and should be better described/elaborated. This paragraph seems to be drawing conclusions, and then explaining an experiment (again slightly too briefly) that these conclusions are based on, so cause/effect and relationship to previously described results is obscured. I assume this is another test independent test of the model — but what stimulus was being used, and why was adaptation not required? Also, I thought that — once trained — all the models were tested on fixed parameter values. So what were these fixed on? (was it not trained)?

Before training the photoreceptor-CNN model, initial parameters have to be assigned to all the parameters of the model. In the case of CNN layers, the parameters (weights) are randomly initialized from an i.i.d. uniform distribution. For the photoreceptor layer, parameters can be initialized to either reflect a rod photoreceptor or cone photoreceptor using experimentally derived values from rod and cone photoreceptors. Here we initialize it to reflect rod photoreceptors. During the training phase, all the CNN parameters are learnt, and we allow for some photoreceptor parameters to be learnt as well. These trainable parameters consisted of the photopigment decay rate, the phosphodiesterase (PDE) activation rate, the PDE decay rate, the rate of Calcium extrusion and sensitivity to inputs. Once the model is trained, the learnt parameters may deviate from those of true biological rods.

In this last experiment described (Fig. 3c) all parameters of the photoreceptor model are set to be non-trainable except the parameter that controls its sensitivity to inputs. Once the photoreceptor-CNN model is trained, the photoreceptor layer parameters that were initialized to reflect rods, still reflect true biological rods.

Adaptation is achieved irrespective of whether the photoreceptor model is trainable or not. It results from the nonlinear processes – including feedback – within the biophysical model. Different parameters for the model may lead to different dynamics. For example, the same model can be used for both rods and cones by changing its parameters to reflect either photoreceptor type.

In all experiments in this section the underlying training and evaluation scheme was the same. Models were first trained on RGC responses to white noise movies and then fine tuned on responses to naturalistic stimuli.

We have now updated this paragraph in the main text to describe this better. Please see lines 281-297.

SMALLER EXPLANATIONS/CLARIFICATIONS

1. The description of cell criteria was added to the Results (lines 78-9) but "reliability across experimental conditions" was vague and only explained in detail in the methods. It would clarify to simply say that "reliability" here is not a general/ambiguous term but refers to detailed criteria explained in the methods.

We now add a reference to the Methods section in the relevant lines of the main text.

2. Line 86 about batchnorm: This is now very well described in the methods, but the sentence here in the text gives the wrong impression, I believe. As stated in the methods, this is a running average during training only but that results in fixed input-output for evaluation and prediction purposes, and does not involve dynamic z-scoring like the LayerNorm. As written, this seems to imply that it is a dynamic "batch-by-batch" z-scoring, which should be clarified.

We have now clarified this and added a reference to the Methods section where more details are provided.

3. lines 93-93: the LayerNorm layer: it seems to say that the mean across all time points in each movie (80 frames) was removed on a pixel-by-pixel basis, which seems that it would disrupt/cause spatial contrast where there was not any. Am I understanding this right, and would this be a problem? I believe this is probably fine, but could be better described.

You are correct that this process involves normalizing each pixel across the temporal sequence, which can indeed influence the spatial contrast characteristics of each frame. This method was chosen to enhance the model's ability to detect and adjust to temporal variations, which is crucial for the analysis conducted. We have updated the description in the revised manuscript accordingly.

4. Line 121: Why were photoreceptors configured for rods for the whole experiment? Is this just for the experiment with the animated natural images? Why was this the case? Was this the one demonstration (additional experiment) that was performed or was it necessary for the model to work at any luminance. Please also specify whether this rod configuration was indeed for all light levels, or just the first section

The primate retina experiments (first 4 result sections) were performed at relatively low light levels where rod photoreceptors are primarily driving RGC responses. Therefore,

when modeling RGCs from these experiments using the photoreceptor-CNN model, the photoreceptor layer parameters were initialized to represent rod photoreceptors. In the last result section we model rat retina responses at both bright and dark light levels where either the cones or rods are dominant, respectively. To model responses at bright light levels, we initialize the photoreceptor layer parameters to reflect the cone photoreceptors. To model responses at dark light levels, we initialize the photoreceptor layer parameters to reflect rod photoreceptors. In mesopic conditions – which we did not include in this study – we would need multiple channels of the photoreceptor models, to reflect both rod and cone photoreceptors.

5. Line 131: What parameters about the photoreceptor model were learned (line 132)? Not asking at the level of methods, but what did the learned parameters control: "i.e. time scales, the relative amount of cGAMP pool, etc?): can this be easily summarized?
We have included a detailed description of both the trainable and non-trainable parameters in the Methods section and have now added a pointer to it in the main text.

Reviewer #2

The authors have adequately addressed all my comments and substantially improved the manuscript's content. I have no further comments.

We thank the reviewer for their feedback.

Reviewer #3

Thanks to the authors for their comprehensive responses to the reviewers' comments. All of my concerns have been addressed in the revised manuscript. I recommend this paper for publication.

We thank the reviewer for their feedback.

Reviewer #4

The authors thoroughly addressed all of my comments. My main concerns were fairly well solved by adding a new experiment with natural images dynamically shifted using saccade movements.

The new data, figures, organisation and better writing of the manuscript profoundly increased its quality. I believe that this article will provoke the community into questioning the need to include biophysical modeling into CNN architectures for vision. However, much work lies in the

future in order for these models to perform better and to add more precise insights into retinal physiology.

We appreciate these supportive remarks and thank the reviewer for providing constructive feedback.

Some final minor remarks:

1. Line 154: it is not clear how a grid search can decrease more than 300,000 trainable parameters. Are the authors focusing on a specific layer parameters?

We focus on optimizing the number of filters required and their sizes in each layer. Changing the number of filters required can have a huge impact on the number of parameters. For example, having 15 filters in the last layer of a CNN model instead of 20 filters (7x7 size) can reduce the parameter count by 200,000. This is because the output of each 7x7 filter in the third layer is a 20x33 array containing the filter outputs at each spatial location. Those then get flattened and passed through a dense layer to generate the predicted RGC responses. In that layer, each filter output and spatial location has its own weight with which it affects each RGCs activation. The block of that weight matrix reflecting these extra 5 filters thus has $(20 \times 33) \times 5 \times 57 \sim 188,100$ weights in it, each of which are trainable parameters. In addition, there are the 57 trainable parameters defining the biases of the 57 output units.

2. Figure 3b: is there a difference between ON and OFF cells? It seems that a linear photoreceptor model may be good for many OFF cells.

It is indeed possible that there are differences in performance of ON and OFF cells as they might rectify the outputs with different slopes and degrees of saturation. However, we do not investigate these differences in the current manuscript and leave them for future studies.

3. Figure 2: Since an interesting claim is made about the fixed photoreceptor model CNN (Fig 3c), it may be nice to see an example of its behavior in a Figure 2c.

We have now included the example RGC's predicted response using the fixed photoreceptor-CNN as Figure 2c.

4. Line 219: Could the authors make a rationale on why they obtain around 40% more of FEV for white noise model (88%) vs the natural movies (49%)? Do mid-gate cells have a higher FEV, and then this explains the difference? (different balance of cells types are included in each dataset). Is the model proposed not enough to predict as nicely the

natural movies? Is there any improvement the authors foresee will point in the direction of achieving better results, as the 88% reported from Figure 4?

The lower performance with naturalistic movies is most likely attributable to the disparity in quantities of training data (8 minutes of naturalistic movies versus 40 minutes of white noise movies), but other factors may also play a role. For example the network lacks other adaptive mechanisms in the retina found downstream of the photoreceptors, such as spike frequency adaptation in RGCs, which may be required to capture the wider range of contrasts and temporal correlations that are present in naturalistic movies but absent in white noise stimuli. We have now included this explanation in Discussion (lines 574-582).

Introducing adaptive recurrent units (developed by Geadah et al., 2022) to the output layer of the photoreceptor-CNN, which implement spike frequency adaptation through dynamic control of a nonlinearity, is a potential solution. ARUs at the output layer would also enable the network to have output units with diverse properties, similar to the diversity of RGCs and may help improve performance across multiple RGC types.

5. Line237: “We attribute this to the model’s ability to modulate output properties based on mean light level”. This claim, although it seems very plausible, is not properly explained with data. What property are the authors talking about? A detailed list of the fitted parameters of the model is now given, and there is a parameter changing 2 orders of magnitude. Is this one the main responsible for the success of the PR model across light levels? I think this is a missed opportunity, as they say in line 373, that other models could be used to obtain similar results.

The photoreceptor model is able to modulate output properties such as the response gain and kinetics. We have now made this explicit in the main text.

6. Line 470: “This was necessary as we first trained the models on checkerboard movie and then fine-tuned the same model with naturalistic movies”. This is a technical reason, but not necessary, they could have upsampled the checkerboard movie.

To train the model efficiently requires unique training examples which cannot be achieved by upsampling the stimuli as this just repeats the same set of frames. We therefore first trained the models using white noise stimuli for which we had 36 minutes of data and then fine tuned the model using the limited (6 seconds x 8 unique training movies x 10 repeats = 8 minutes) naturalistic movie data that we had.